



# Decrease of the European NO$_x$ anthropogenic emissions between 2005 and 2019 as seen from the OMI and TROPOMI NO$_2$ satellite observations

Audrey Fortems-Cheiney[1,*], Gregoire Broquet[1], Robin Plauchu[1], Elise Potier[1,*], Antoine Berchet[1],
Isabelle Pison[1], Adrien Martinez[1], Rimal Abeed[1], Gaelle Dufour[2], Adriana Coman[3], Dilek Savas[2],
Guillaume Siour[3], Henk Eskes[4], Hugo Denier van der Gon[5], and Stijn Dellaert[5]

[1]Laboratoire des Sciences du Climat et de l'Environnement, CEA-CNRS-UVSQ, Gif-sur-Yvette, France
[2]Université Paris Cité and Univ Paris Est Créteil, CNRS, LISA, F-75013 Paris, France
[3]Univ Paris Est Créteil and Université Paris Cité, CNRS, LISA, F-94010 Créteil, France
[4]Royal Netherlands Meteorological Institute (KNMI), De Bilt, the Netherlands
[5]Department Air Quality and Emissions Research, TNO, P.O. Box 80015, 3508 TA Utrecht, Netherlands
[*]now in Science Partners, Quai de Jemmapes, 75010 Paris, France

**Correspondence to:** Audrey Fortems-Cheiney (audrey.fortems@science-partners.com)

**Abstract.** There are great expectations about the detection and the quantification of NO$_x$ emissions using NO$_2$ tropospheric columns from satellite observations and inverse systems. This study assesses the potential of the OMI-QA4ECV and TROPOMI satellite observations to improve the knowledge on European NO$_x$ emissions at the regional scale and to inform about the spatio-temporal variability of NO$_x$ anthropogenic emissions from 2005 to 2019, at the resolution of 0.5° over Europe. We first characterize the level of consistency between retrievals from OMI-QA4ECV and from the more recent reprocessing of the TROPOMI data, called TROPOMI-RPRO-v2.4, and the implications of the possible inconsistencies for inversions. Furthermore, starting from European emission estimates from the TNO-GHGco-v3 inventory for the year 2005, regional inversions using the Community Inversion Framework coupled to the CHIMERE chemistry-transport model and assimilating satellite NO$_2$ tropospheric columns from OMI and TROPOMI have been performed to estimate the European annual and seasonal budgets for the year 2019. Both the OMI and TROPOMI inversions show decreases in European NO$_x$ anthropogenic emission budgets between 2005 and 2019. Nevertheless, the magnitude of the reductions of the NO$_x$ anthropogenic emissions are different with OMI and TROPOMI data, with decreases in EU-27+UK between 2005 and 2019 of 16% and 45%, respectively. A TROPOMI inversion giving more weight to the satellite data becomes consistent with the independent TNO-GHGco-v3 inventory for the year 2019, with annual budgets for EU-27+UK showing absolute relative difference of only 4%. These TROPOMI inversions are therefore in agreement with the magnitude of the decline in NO$_x$ emissions declared by countries, when aggregated at the European scale. However, our results —with OMI and TROPOMI data leading to different magnitudes of corrections on NO$_x$ anthropogenic emissions—suggest that more observational constraints would be required to sharpen the European emission estimates.



## 1 Introduction

Air pollution is still a major health concern in Europe (EEA, 2023). Particularly, up to now all European Union (EU) countries report levels of nitrogen dioxide ($NO_2$) above the 2021 World Health Organization (WHO) global air quality guidelines, and around 64,000 premature deaths are caused by $NO_2$ pollution each year (EEA, 2023). In Europe, the main sources of $NO_2$ are road transport —responsible for 39% of the emissions in 2019 (EEA, 2022), thermal power plants and industrial activities

(EEA, 2023). $NO_2$ is mainly produced in the atmosphere by the oxidation of nitric oxide (NO), which is emitted by the same activities. To reduce $NO_2$ concentrations to levels that do not impact human health, nitrogen oxides ($NO_x$=NO+$NO_2$) emissions need to be cut down.

The EU has comprehensive regulations governing air quality. The key legal instrument in this regard is the Ambient Air Quality Directive (2008/50/EC). In addition, there are other EU regulations and directives that address specific aspects of air quality,

such as the National Emission Ceilings (NEC) Directive (2016/2284/EU) (OJE, 2016), which sets expected reductions of $NO_x$ emissions compared with 2005, often considered as the baseline year of regulation policies. To design efficient regulatory policies and assess their effectiveness, an accurate account of emissions in space and time is essential.

Budgets and source sector distributions for air pollutants often come from bottom-up (BU) inventories, based on the statistics of socio-economic activities and fuel consumption and on emission factors per activity type. However, the quantification of

anthropogenic $NO_x$ emissions following a BU approach suffers from relatively large uncertainties, especially those coming from the emission factors, and from the use of average values for these factors despite their large spatial and temporal variability. For instance, the European Monitoring and Evaluation Programme (EMEP) inventory reports 50-200% uncertainties in their sectoral budgets of $NO_x$ anthropogenic emissions at the national and annual scales (Kuenen and Dore, 2019). According to Schindlbacher et al. (2021), the uncertainty estimates for national anthropogenic $NO_x$ emissions range from 5% to 56% in

Europe. The spatial distribution and temporal variability of mapped emission inventories often rely on proxies and typical temporal profiles, which inevitably introduce errors in the spatial and temporal variations of the corresponding series of emission maps.

Since the year 2000, space-borne instruments are able to monitor atmospheric concentrations of nitrogen dioxide ($NO_2$). Among these instruments are the Global Ozone Monitoring Experiment (GOME) (Burrows et al., 1999), GOME-2 (Munro

et al., 2016), the SCanning Imaging Absorption spectroMeter for Atmospheric CHartographY (SCIAMACHY) (Burrows et al., 1995; Bovensmann et al., 1999), as well as the Ozone Monitoring Instrument (OMI) (Levelt et al., 2006, 2018). These instruments offer a global daily coverage with spatial horizontal resolutions of several hundreds of squared-kilometers. With archives covering more than 20 years, satellite data have provided valuable information for monitoring the chemical composition of the atmosphere, and for documenting the long-term variations of $NO_2$ concentrations associated to economic changes or pollution

control legislations during the last decade (van der A et al., 2008; Castellanos and Boersma, 2012; Schneider et al., 2015; Lamsal et al., 2015; Krotkov et al., 2016; Miyazaki et al., 2017; Li and Wang, 2019; Georgoulias et al., 2019; Silvern et al., 2019; Fortems-Cheiney et al., 2021a; van der A et al., 2024).

Due to the growing availability and reliability of satellite observations, progress has been made to develop atmospheric trans-





port inverse modelling for the estimation of $NO_x$ emissions, also referred to as top-down (TD) or inverse modelling. The available $NO_2$ satellite observations have been widely used to estimate anthropogenic and/or biogenic $NO_x$ emissions, mainly through comparisons between the observations of $NO_2$ tropospheric vertical column densities (TVCDs) and their simulations with atmospheric chemistry transport models, the latter being fed with spatially distributed emissions from BU inventories and land-surface models. The retrieval by atmospheric inversion of robust anthropogenic and/or biogenic emissions assumes the discrepancies between simulated and observed TVCDs to be mainly due to these BU emission estimates. In reality, the discrepancies between simulated and observed TVCDs are due to a set of different causes, which can all interact. In particular, the discrepancies can be caused by the limitations of the modelling of the atmospheric transport and chemistry of $NO_x$ (Stavrakou et al., 2013). Part of the discrepancies are also related to the observations, with instrumental errors and retrieval biases (Lorente et al., 2017). In addition, the retrieval of quantitative tropospheric column amounts with their corresponding averaging kernels, which account for the vertical sensitivity of the satellite measurements, is complex and remains a major challenge. The retrieved TVCDs and their vertical sensitivity indeed strongly depend on different factors (e.g., cloud fraction and effective cloud pressure, the assumed prior profile shape, surface albedo, the presence of a stratospheric background and aerosols) of the modelling and inversion of the radiative transfer in the atmosphere (Boersma et al., 2004; van Geffen et al., 2022a). Therefore, satellite data inter-comparison studies reveal large differences between different retrievals of the same compound from the same measurements of a given instrument based on different algorithms. For instance, the magnitude of $NO_2$ TVCDs from two global OMI retrieval products (i.e., OMI-NASAv3 from the National Aeronautics and Space Administration (NASA) and OMI-DOMINOv2 from the Royal Netherlands Meteorological Institute (KNMI)) differ by 50% over densely populated areas in China, and they show different trends over the past decade at the regional scale (Zheng et al., 2014; Qu et al., 2017; Lorente et al., 2017). Qu et al. (2020) compared three OMI $NO_2$ retrieval products (i.e., OMI-NASAv3, OMI-DOMINOv2 and the Quality Assurance for Essential Climate Variables product OMI-QA4ECV), and showed that the different vertical sensitivities in the $NO_2$ retrievals explain for a large part the discrepancies between the retrieved $NO_2$ TVCDs.

Global scale TD approaches have been first considered to derive the spatio-temporal variability of the $NO_x$ emissions over large regions (Martin et al., 2003; Boersma et al., 2008; Stavrakou et al., 2008; Lamsal et al., 2011; Miyazaki et al., 2017). However, there are scientific and societal needs for the quantification and mapping of pollutant emissions at a relatively high-spatial resolution, which requires the use of regional-scale inversion systems. Regional systems based on mesoscale chemistry transport models are indeed suited to the simulation and analysis of the $NO_2$ concentrations, with a spatial resolution of several dozen kilometers (Mijling and van der A, 2012; Mijling et al., 2013; Lin, 2012; Ding et al., 2017; Visser et al., 2019; Fortems-Cheiney et al., 2021b; Savas et al., 2023; van der A et al., 2024), and even down to a resolution of 10 km×10 km (Plauchu et al., 2024).

Beyond the spatial resolution of the chemistry-transport models (CTM), inversion approaches need to manage the high dimensionality of the problem for the estimation of $NO_x$ budgets at relatively fine spatio-temporal scales, typically at the monthly and at the national to sub-national scales in Europe. Regional CTMs coupled to such approaches can be computationally expensive. To mitigate the computational cost, mass balance approaches have been employed at the regional scale (Visser et al., 2019). These approaches account for the non-linear relationships between $NO_x$ emission changes and $NO_2$ TVCDs via reactions with



hydroxyl radicals (OH) but with simple scaling factors. A more robust account for the complex $NO_x$ chemistry is required for

the accurate derivation of $NO_x$ emissions from $NO_2$ satellite data (Stavrakou et al., 2013; Ding et al., 2017; van der A et al., 2024). In addition, in many cases, the errors from both the CTM and retrievals are ignored and the emissions are scaled to get an optimal fit to the observations. Taking the errors from both the model and retrievals into account is, nevertheless, considered as essential for improving emission estimates (Miyazaki et al., 2017). In this context, complex Bayesian approaches with ensemble Kalman filter inverse modeling techniques or variational approaches may have a key-role to play, since they are

designed to take into account i) the non-linearities of the $NO_x$ chemistry and ii) errors from both the model and retrievals.

Since 2017, the TROPOspheric Monitoring Instrument (TROPOMI, Veefkind et al. (2012)) onboard the Copernicus Sentinel-5 Precursor (S5P) has been providing $NO_2$ TVCDs images (over a wide swath), at higher spatial resolutions and improved signal-to-noise ratio, compared to previous missions such as OMI (van Geffen et al., 2020). With variational inversions, Plauchu et al. (2024) assessed the potential of the TROPOMI observations to inform about $NO_x$ emissions in France from 2019 to 2021 at

the national to urban scales. Their results open positive perspectives regarding the ability of inversions to support the validation or improvement of inventories with TROPOMI observations, at least at the local level for emission hot spots generating a relatively strong local signal, as these are better caught and exploited by the inversions than the larger scale signals (Plauchu et al., 2024). While previous satellite instruments could already provide information over emission hot spots (e.g, over large urbans areas in Europe (Fortems-Cheiney et al., 2021a)), TROPOMI might be now more relevant than these previous satellite

instruments to monitor anthropogenic emissions (Zheng et al., 2020; Li et al., 2023).

Among the instruments providing a long archive of $NO_2$ observations, OMI has the highest spatial resolution and least degradation through time (Levelt et al., 2018; Schenkeveld et al., 2017). OMI and TROPOMI present similarities: the retrievals from TROPOMI and from OMI-Q4ECV are based on similar algorithms, using the same prior profiles and assimilation technique to estimate the stratospheric column from the global model TM5-MP, and they make measurements at nadir at about the same

local time. Therefore, we expect these data sets to be consistent, with similar $NO_2$ TVCDs over horizontal and temporal scales larger than a few hundred kilometers resolution and the month. Nevertheless, a comparison of TROPOMI version 1.3 $NO_2$ TVCDs to OMI-QA4ECV observations shows differences in terms of monthly $NO_2$ average varying from a few % up to -40% over polluted regions (Western Europe, Eastern China, Eastern USA, Middle East), the largest differences occuring in winter (Lambert et al., 2021). Actually, one of the differences between the two retrievals is the cloud pressure retrieval, which could

have large impacts on the results. In particular, the Fast Retrieval Scheme for Clouds from Oxygen absorption band FRESCO-S version implemented for TROPOMI in v1.0 to v1.3 has been known to overestimate the cloud pressure, leading to a high bias in the air-mass factors and a low bias in the tropospheric columns (Lambert et al., 2021). Comparisons with ground-based measurements have also shown that versions v1.2 and v1.3 of TROPOMI data lead to $NO_2$ TVCDs that are too low by 22% to 37% for clean and slightly polluted scenes, and up to 51% over highly polluted areas (Verhoelst et al., 2021). Efforts have

been made to correct such biases in the recent versions of the TROPOMI data, and seem to lead to a better agreement with OMI-QA4ECV observations and with ground-based measurements (Lambert et al., 2021; van Geffen et al., 2022b, a).

In this context, we first characterize the level of consistency between retrievals from OMI-QA4ECV and from the more recent reprocessing of the TROPOMI data, called TROPOMI-RPRO-v2.4, and the implication of the possible inconsistencies



for inversions. Regional simulations of NO$_2$ TVCDs in Europe with the CHIMERE CTM (Menut et al., 2013; Mailler et al.,
2017) are used to indirectly analyze the consistency between OMI and TROPOMI products via the differences between these
NO$_2$ TVCDs and the ones simulated by CHIMERE. Furthermore, this study assesses the potential of the OMI and TROPOMI
satellite observations to inform about the evolution of NO$_x$ anthropogenic emissions between year 2005 and year 2019 at the
regional to national scales in Europe. We have performed 1-year inversions using i) an estimate of the NO$_x$ anthropogenic
emissions from the TNO-GHGco-v3 inventory (Dellaert et al., 2021) for the year 2005 to define the prior estimate that the
variational inversion framework corrects to better fit the observations and ii) OMI and TROPOMI observations for year 2019
to estimate European NO$_x$ emissions for the year 2019. According to BU inventories, NO$_x$ anthropogenic emissions have
been strongly decreased –by about 36%– in Europe since 2005 (EEA, 2021) which is the year corresponding to the inventory
used, by construction of our experimental framework, as the inversion prior estimate for 2019. The evaluation of the emissions
estimated from the inversions with comparisons to the NO$_x$ emission estimates from the TNO-GHGco-v3 inventory for the
year 2019 should thus provide insights on the current capability of inversion assimilating satellite NO$_2$ observations to quantify
the evolution of the European and national NO$_x$ budgets between 2005 and 2019.

For these objectives, we have used a European scale variational inverse modeling system based on the coupling between
the variational mode of the Community Inversion Framework (CIF) (Berchet et al., 2021) and CHIMERE and its adjoint code
(Fortems-Cheiney et al., 2021b).
We first describe the CHIMERE configuration for Europe, the NO$_2$ satellite observations, and the variational inversion
method in Section 2. Section 3 presents our results, including the comparison between the CHIMERE simulations, the OMI
and TROPOMI NO$_2$ TVCDs and the posterior estimates of NO$_x$ European anthropogenic emissions.

## 2   Data and Methods

### 2.1   The CIF-CHIMERE inversion system

The Community Inversion Framework (CIF) is a modular inverse modeling platform which can drive various data assimilation
schemes and various CTMs (Berchet et al., 2021). Here, the CIF drives the CHIMERE CTM (Menut et al., 2013; Mailler et al.,
2017) and its adjoint code (Fortems-Cheiney et al., 2021b). The coupling between the CIF and CHIMERE and its adjoint code
and their use for variational inversions takes advantage of the developments in the Bayesian variational atmospheric inversion
system PYVAR-CHIMERE to account for reactive species (Fortems-Cheiney et al., 2021b).

### 150   2.2   Prior estimates of the NO$_x$ emissions in Europe

The principle of the inversion is to correct *a priori* emission maps later on referred as "prior" emissions. The prior estimates of
NO$_x$ emissions in this study are based on anthropogenic NO$_x$ emission estimates from the TNO-GHGco-v3 gridded inventory
(Dellaert et al., 2021) for the year 2005. We have chosen such an estimate for year 2005 to assess the potential of satellite
observations for year 2019 to quantify the high spatio-temporal differences in the NO$_x$ emissions between 2005 and 2019. The



$NO_x$ emission estimates from the TNO-GHGco-v3 inventory for the year 2019 is used to evaluate the inversion results. We have also performed inversions i) both using $NO_x$ prior emissions and assimilating satellite observations for the year 2005 and ii) both using $NO_x$ prior emissions and assimilating satellite observations for the year 2019 (Table 1) to assess the impact of the prior inventory on the $NO_x$ emissions estimated from the inversions.

The TNO-GHGco version is an update of the TNO inventory documented in Kuenen et al. (2014) and in Super et al.
(2020), based on country emission reporting to the European Monitoring and Evaluation Program (EMEP)/Center on Emission Inventories Projection (CEIP). The TNO-GHGco-v3 inventory maps $NO_x$ emissions at a 6×6 km$^2$ horizontal resolution. It combines emissions from area sources, set at the surface, and from point sources. Emissions from point sources, mainly from the energy production and the industrial sectors, are distributed on the first eight vertical model layers in CHIMERE depending on the typical injection heights provided in the TNO inventory, based on Bieser et al. (2011).

In the TNO inventory, annual and national budgets are disaggregated in space based on proxies of the different sectors (Kuenen et al., 2014). Temporal disaggregation is based on temporal profiles provided per GNFR sector code with typical month to month, weekday to week-end and diurnal variations (Ebel et al., 1994, Menut et al., 2011). Following the GENEMIS recommendations (Kurtenbach et al., 2001; Aumont et al., 2003), we have speciated the TNO-GHGco-v3 $NO_x$ emissions as 90% of NO, 9.2% of $NO_2$, and 0.8% of nitrous acid (HONO) emissions.

CHIMERE is fed with NO biogenic soil emissions from the Model of Emissions of Gases and Aerosols from Nature (MEGAN) model for the year 2019 (Guenther et al., 2006), with a ~1×1 km$^2$ spatial resolution. MEGAN does not take the impact of agricultural practices into account, even though it covers both natural and agricultural areas. There are large uncertainties in the $NO_x$ emissions due to agriculture, and in principle, there could be some overlapping between the agricultural and purely natural soil $NO_x$ emission estimates. It explains why these emissions are not provided by the TNO inventory. There-
fore, we do not include a specific agricultural soil $NO_x$ emissions component in our prior estimation of the $NO_x$ emissions. The lightning $NO_x$ fluxes, whose impact on $NO_2$ concentrations is very small in Europe even in summer (Menut et al., 2020), are not accounted for. Fire emissions are also ignored, as their contribution to the $NO_x$ total emissions and to the long-term $NO_2$ concentration trends over Europe is small. The anthropogenic emissions for volatile organic compounds (VOCs) are obtained from the EMEP inventory (Vestreng et al., 2005).

The different emission products have been aggregated at the 0.5°×0.5° horizontal resolution of the CHIMERE grid. The maps of monthly budgets for total, anthropogenic and natural $NO_x$ emissions at 0.5° resolution are shown in Figure 1 for August 2019. Even in summer when the biogenic $NO_x$ emissions are high, the anthropogenic emissions contribute to about 95% of the total $NO_x$ emissions in Europe with a budget of about 815 kteqNO$_2$. This unit means that the mass of both NO and $NO_2$ emissions are calculated using the $NO_2$ molar mass.

**2.3 Configuration of the CHIMERE CTM for the prior simulation of NO$_2$ concentrations in Europe**

We present the configuration of the CHIMERE CTM used in this study and the specific elements of the simulation, which will be used as the prior of the variational inversions. Here, the configurations of CHIMERE and of its adjoint code are driven by the CIF to simulate $NO_2$ atmospheric concentrations over Europe over a 0.5°×0.5° regular horizontal grid with 17





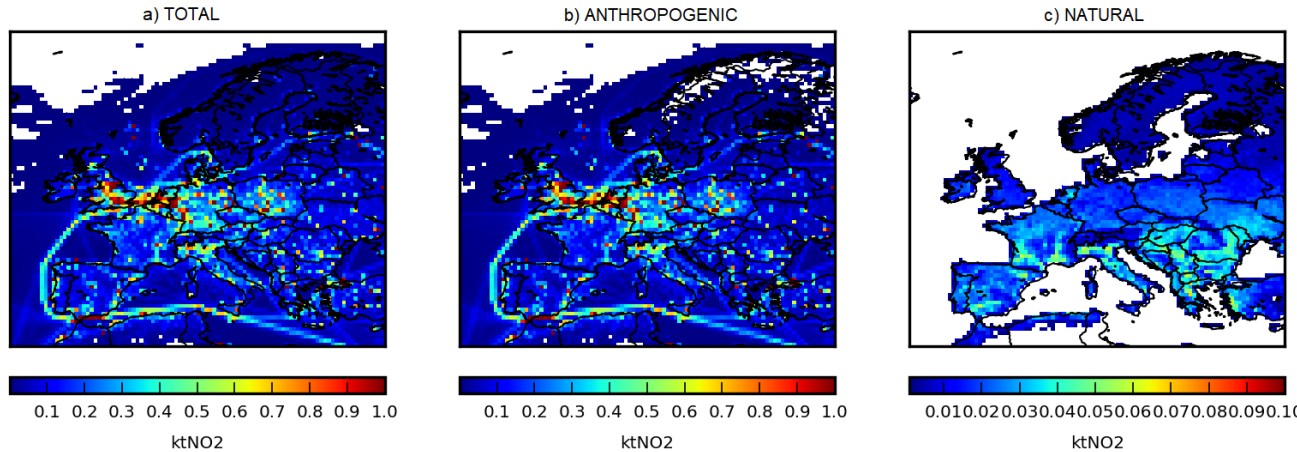

**Figure 1.** *Monthly budget of prior NO$_x$ emissions corresponding to the a) total, b) anthropogenic from the TNO-GHGco-v3 inventory and c) biogenic from the MEGAN model, per model grid-cell, in kteqNO$_2$, in August 2019. Note that the scale is different for anthropogenic and biogenic emissions.*

vertical layers, from the surface to 200 hPa, with 8 layers within the first two kilometres. The domain covers 15.25°W-35.75°E; 31.75°N-74.25°N and includes 101 (longitude) x 85 (latitude) x 17 (vertical levels) grid-cells. The chemical scheme used here is MELCHIOR-2, with more than 100 reactions, including 24 for inorganic chemistry (Lattuati, 1997; Derognat et al., 2003). Due to the need for a compromise between the robustness of the simulation of the chemistry in the model and the computational cost with a complex chemical scheme, the aerosol modules of CHIMERE have not been included in its adjoint code yet and are therefore not activated in the CHIMERE forward simulations, as explained in Fortems-Cheiney et al. (2021b).

CHIMERE is driven here by the European Centre for Medium-Range Weather Forecasts (ECMWF) operational meteorological forecast for the year 2019 (Owens and Hewson, 2018). Considering NO$_2$ short lifetime, we do not consider its import from outside the domain. Nevertheless, the lateral and top boundaries for longer lived species such as ozone (O$_3$), nitric acid (HNO$_3$), peroxyacetyl nitrate (PAN), which participate to the NO$_x$ chemistry, are taken into account (Fortems-Cheiney et al., 2021b). Climatological values from the LMDZ-INCA global model (Szopa, 2008) are used to prescribe the concentrations of these species at the model boundaries.

### 2.4 Satellite Observations

As significant biases between the OMI and TROPOMI tropospheric columns have been described in previous studies in winter (Lambert et al., 2021; Verhoelst et al., 2021; van Geffen et al., 2022b), the month of January 2019 is being particularly used to compare these observations in the following.





### 2.4.1 OMI-QA4ECV-v1.1

The Ozone Monitoring Instrument (OMI) is an ultraviolet-visible (UV-Vis) instrument launched in July 2004 onboard the Earth Observation System (EOS) Aura satellite, which flies on a 705 km sun-synchronous orbit that crosses the Equator at approximately 13:40 LT. The nominal footprint of the OMI ground pixels is $24\times13$ km$^2$ (across$\times$along track) at nadir. With a swath of about 2600 km, it provided daily global coverage for NO$_2$ (no longer achieved due to the row anomaly). OMI uses the O$_2$-O$_2$ absorption feature for the cloud pressure retrieval (Veefkind et al., 2016).

The NO$_x$ inversions described in Section 2.5 assimilate NO$_2$ TVCDs from the OMI-QA4ECV-v1.1 (www.qa4ecv.eu and http://temis.nl/qa4ecv/no2.html, Boersma et al., 2017, 2018). The data selection follows the criteria of the data quality statement (Boersma et al., 2017): the processing error flag equals 0, the solar zenith angle is lower than 80°, the snow ice flag is lower than 10 or equal to 255, the ratio of tropospheric air mass factor (AMF) over geometric AMF is higher than 0.2 to avoid situations in which the retrieval is based on very low (relative) tropospheric air mass factors and the cloud fraction is lower than 0.5. We use an additional criterion for the selection of the observation to be assimilated: the error associated to the retrieval must be lower than 100%. Note that the OMI-QA4ECV-v1.1 reprocessed data sets officially covers the period 2005-2018. This processing is using ERA-Interim 60 layer meteorological reanalyses from the ECMWF as driver. To facilitate comparisons with TROPOMI, the OMI dataset was further extended to 2019 using 137 layer ECMWF operational meteorological data. For 2019, the pressure levels are identical for the OMI and TROPOMI products.

### 2.4.2 TROPOMI RPRO-v2.4

The Tropospheric Monitoring Instrument (TROPOMI, (Veefkind et al., 2012)) was launched onboard the Copernicus Sentinel-5 Precursor (S5P) satellite in October 2017. It flies on a 824 km altitude sun-synchronous orbit that crosses the Equator at approximately 13:40 LT. This imaging spectrometer covers a UV-Vis band supporting the derivation of NO$_2$ TVCDs observations. The nominal footprint of the TROPOMI ground pixels is of about $7\times3.5$ km$^2$ before 6 August 2019 and $5.5\times3.5$ km$^2$ after 6 August 2019 at nadir. With a swath of about 2600 km on ground, it provides daily global coverage for NO$_2$. TROPOMI uses the Fast Retrieval Scheme for Clouds from Oxygen absorption band (FRESCO) retrieving cloud information from the O$_2$ A-band (Lambert et al., 2021). In addition, TROPOMI-v2.4 uses the TROPOMI dependent Lambertian-equivalent reflectivity (DLER) v1.0 surface albedo dataset (Tilstra et al., 2024), in contrast to OMI which makes use of the OMI Lambertian equivalent reflectance (LER) (Kleipool et al., 2008). Furthermore, TROPOMI applies dynamic albedo adjustments (van Geffen et al., 2022a) which is not done in the OMI-QA4ECV data set.

Our selection of the TROPOMI data to be assimilated in the inversions described in Section 2.5 follows the criteria of van Geffen et al. (2022b). We only select observations with a quality assurance (qa) value higher than 0.75. Like OMI, we only select observations when the error associated to the retrieval is lower than 100%.



### 2.4.3 Comparison between simulated and observed NO$_2$ TVCDs

OMI and TROPOMI have different horizontal resolutions and more generally a different spatio-temporal sampling. The comparison between the data from the two instruments as well as the comparisons to the simulated TVCDs have therefore been done over a common projection, within the $0.5°{\times}0.5°$grid-cells of the CHIMERE CTM.

To make comparisons between simulations and satellite observations, the simulated vertical profiles are first interpolated on the satellite's levels, with a vertical interpolation from CHIMERE's levels. Then, the averaging kernels (AKs) from the OMI and TROPOMI products, respectively, are applied to the simulated profiles to account for the variations of the vertical sensitivity of the satellite retrievals (Eskes and Boersma, 2003; Boersma et al., 2017; van Geffen et al., 2022b). It is important to note that the OMI and the TROPOMI observations are accompanied with differences in their AKs, mainly explained by differences in their cloud pressure retrievals. The lower spatial resolution of OMI compared to TROPOMI also leads to differences in the distribution of the effective cloud fractions, with TROPOMI having relatively more cloud free and fully clouded pixels, which impacts the AKs and their shapes. Over land, OMI and TROPOMI present similar sensitivity near the surface, both over polluted areas defined as areas where the NO$_2$ TVCDs are higher than $2{\times}10^{15}$ molec.cm$^{-2}$ and over rural areas (Figure 2). However, due to lower cloud pressures on average in OMI than TROPOMI and consequently smaller air mass factors (AMFs) in the mid to lower troposphere, the AKs in OMI tend to be smaller than in TROPOMI above the first four levels of the satellites. These different vertical sensitivities and amplitudes of the AKs in the NO$_2$ OMI and TROPOMI retrievals will affect the simulated NO$_2$ TVCDs over Europe.

As the spatial resolution of OMI or TROPOMI data is finer than that of the chosen CTM model grid, the selected OMI or TROPOMI TVCDs are aggregated into "super-observations", as recommended by Rijsdijk et al. (2024). In order to associate the super-observations to an actual AK profile, the super-observations have been taken as the observation (TVCD and AKs) corresponding to the value closest to the mean of the OMI or TROPOMI TVCDs within the $0.5°{\times}0.5°$ model grid-cell and within the CHIMERE physical time step of about 5-10 minutes, as in Plauchu et al. (2024). The choice of the value closest to the mean is different from Fortems-Cheiney et al. (2021b), initially taking the median of the observations for defining super-observations. This choice was made necessary by the high number of TROPOMI observations within the $0.5°{\times}0.5°$ model grid-cell. The number of TROPOMI observations within a TROPOMI super-observation can reach 50, while the number of OMI observations within an OMI super-observation is often lower than five over continental land (Figure 3). We assume that over a set of a few tens of data the mean of the observations is more representative than the median, particularly in grid-cells with high concentrations. The number of TROPOMI and OMI super-observations is given as an example for the month of January 2019, respectively in Figure 4a and in Figure 4b. The TROPOMI super-observations present a coverage higher than the OMI ones by about a factor 4 (about 117,000 against 27,000). The only areas not really covered by TROPOMI are north-eastern Europe because of the snow cover and northern Europe because of the high solar zenith angle (Figure 4a).

The simulated NO$_2$ TVCDs corresponding to the OMI super-observations, using the OMI AKs, are called "CHIMERE-OMI". Similarly, the simulated NO$_2$ TVCDs corresponding to the TROPOMI super-observations, using the TROPOMI AKs, are called "CHIMERE-TROPOMI" in the following.





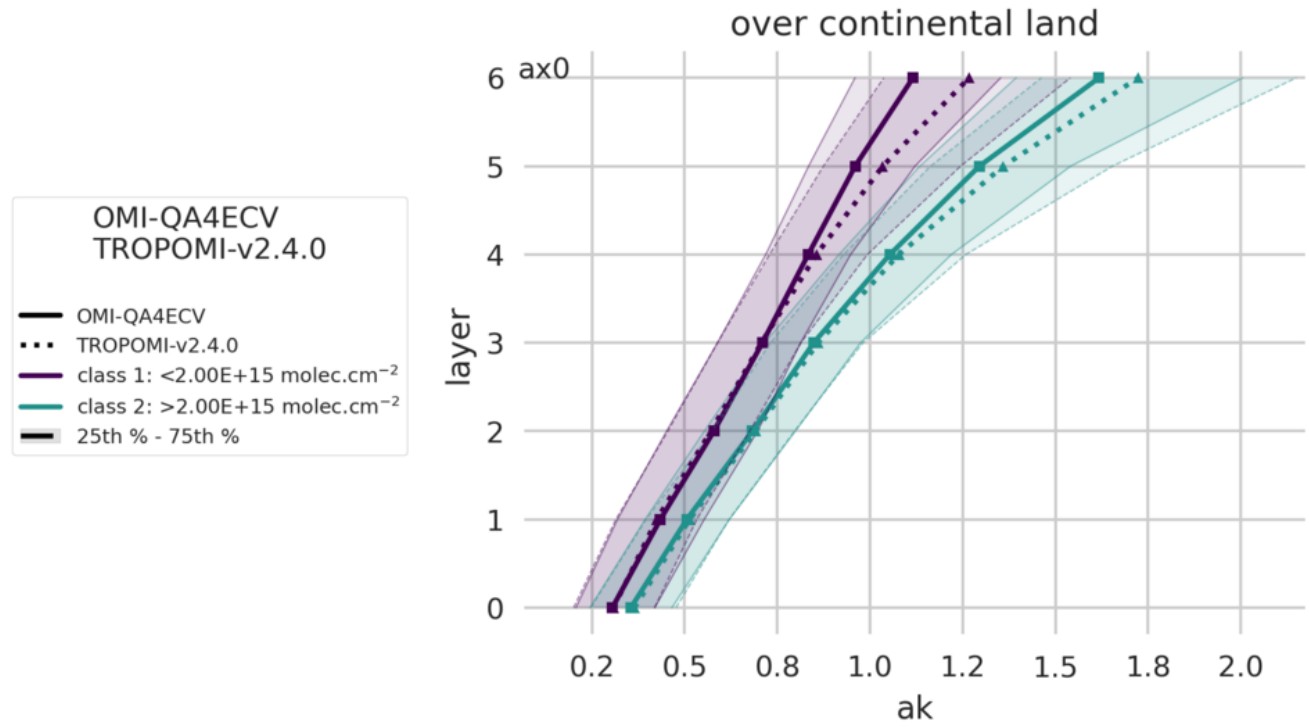

**Figure 2.** Median and quartile profiles of averaging kernels from OMI-QA4ECV and TROPOMI-RPRO-v2.4 super-observations over continental land, for $NO_2$ TVCDs lower and higher than $2 \times 10^{15}$ molec.cm$^{-2}$, for the first six levels of the satellites, in January 2019.

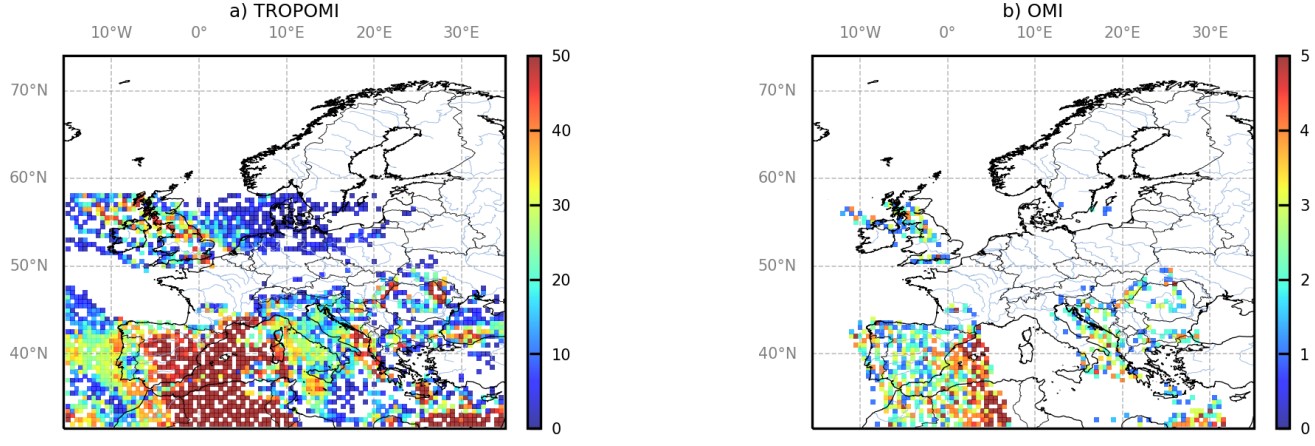

**Figure 3.** Daily number of observations within a) TROPOMI and b) OMI super-observations, for January the 1st in 2019. Note that the scales are different for TROPOMI and OMI.





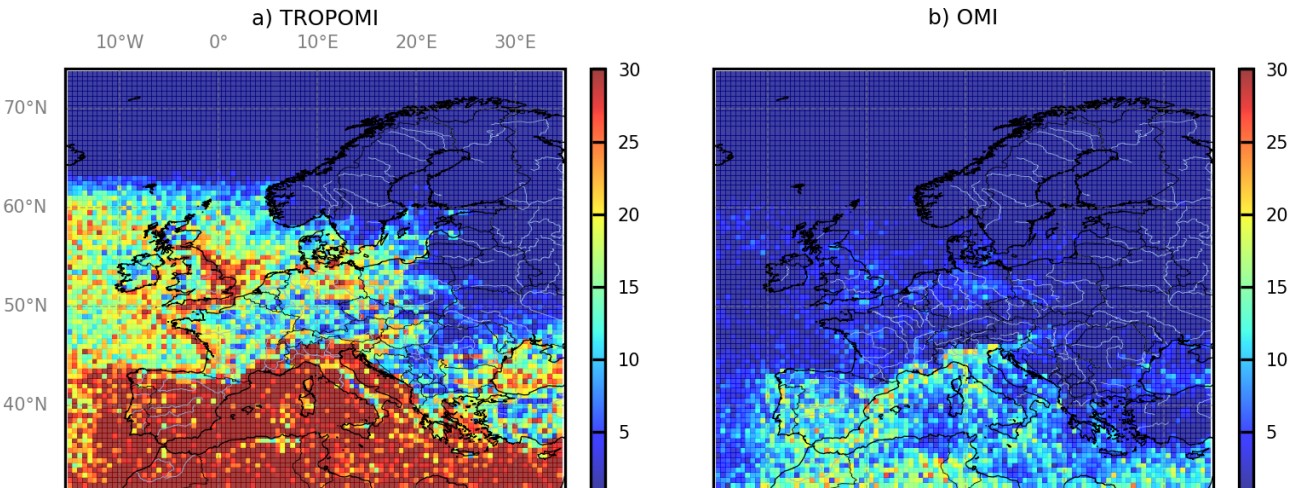

**Figure 4.** Monthly number of a) TROPOMI and b) OMI super-observations, in January 2019.

Various choices can be made for the derivation of the error associated with each super-observation. First, the error associated
270  with each super-observation can be derived from the observation closest to the mean value, as in Plauchu et al. (2024). In this
case, the derivation of the error associated with each super-observation, referred as OMI-A or TROPOMI-A in the following,
is conservative compared to other studies where the super-observation uncertainty is reduced compared to that of individual
observations (Boersma et al., 2016). The reduction of uncertainty when combining several observations accounts for the fact
that the retrieval errors include random noise (in particular, instrumental noise without spatial correlation, i.e. errors which
275  are independent from one observation to the other). However, OMI or TROPOMI $NO_2$ observations probably bear systematic
errors from the instrument and from the retrieval process, which can exhibit important spatial correlations (Rijsdijk et al.,
2024). This can justify a conservative attribution of observation errors to the super-observations.

Nevertheless, in addition to this configuration, we have also performed a sensitivity test, referred to as OMI-B or TROPOMI-B
in the following (Table 1), where the error associated with each super-observation is derived as the error from the observation
280  closest to the mean value multiplied by a factor of $\frac{1}{\sqrt{nbobs}}$ , where $nbobs$ is the number of observations within a super-
observation. In this case, the corresponding super-observation error is smaller than that of the individual observations.

### 2.5  Variational inversion of the $NO_x$ emissions

The reference inversion of $NO_x$ emissions here consists in correcting the prior estimate of the emissions from the TNO-GHGco-
v3 gridded inventory for the year 2005 (presented in Section 2.2) and the initial conditions to improve the fit between $NO_2$
OMI-QA4ECV or TROPOMI-RPRO-v2.4 satellite data for the year 2019 and their simulated equivalents, using a variational
inversion framework similar to that of Plauchu et al. (2024). Series of 1-month inversion windows — independent from each
other — are run and then combined to provide a corrected estimate of $NO_x$ emissions, also called "posterior" in the following,





over the entire year 2019. For each inversion window, the posterior estimate of the emissions is found by iteratively minimizing the cost function $J(\mathbf{x})$:

$$J(\mathbf{x}) = \frac{1}{2}(\mathbf{x} - \mathbf{x}^b)^T \mathbf{B}^{-1}(\mathbf{x} - \mathbf{x}^b) + \frac{1}{2}(\mathcal{H}(\mathbf{x}) - \mathbf{y})^T \mathbf{R}^{-1}(\mathcal{H}(\mathbf{x}) - \mathbf{y})$$

where $\mathbf{x}$, $\mathcal{H}$, $\mathbf{y}$, $\mathbf{B}$, $\mathbf{R}$ are respectively the control vector, the observation operator, the satellite observations, the prior error covariance matrix and the observation error covariance matrix. As in Plauchu et al. (2024), the definition of $\mathbf{x}$ ensures that the inversion solves separately for the two main types of NO emissions: the anthropogenic and the biogenic emissions (without any further sectorization or decomposition into more detailed emission components) and for the anthropogenic $NO_2$ emissions. With such a control vector, the prior $NO/NO_2$ anthropogenic emission ratio speciation from the GENEMIS recommendations (see Section 2.2 is not kept by the inversion. The analysis in the following focuses on the $NO_x$ emissions as the sum of the NO and $NO_2$ emissions.

Contrarily to Fortems-Cheiney et al. (2021b), and as in Plauchu et al. (2024), we characterize the prior uncertainty in both the anthropogenic and biogenic $NO_x$ emissions with log-normal distributions. This allows the inversion system to apply high variations in $NO_x$ emissions while ensuring that the inversion keeps the emissions positive, unlike the classic corrections of the emissions with scaling factors. Details about this feature are given in Plauchu et al. (2024).

Our control vector $\mathbf{x}$ contains:

- the logarithms of the scaling coefficients for NO anthropogenic emissions at a 1-day temporal resolution, at a $0.5° \times 0.5°$ (longitude, latitude) horizontal resolution and over the first 8 vertical levels of CHIMERE i.e, for each of the corresponding $101 \times 85 \times 8$ grid cells,

- the logarithms of the scaling coefficients for $NO_2$ anthropogenic emissions at the same temporal and spatial resolutions as for NO,

- the logarithms of the scaling coefficients for NO biogenic emissions at a 1-day temporal resolution, at a $0.5° \times 0.5°$ (longitude, latitude) resolution and at the surface (over 1 vertical level only), i.e. for each of the corresponding $101 \times 85 \times 1$ grid cells,

- factors scaling the NO and $NO_2$ 3D initial conditions at 0:00 UTC the first day of each month, at a $0.5° \times 0.5°$ (longitude, latitude) resolution and over the 17 vertical levels of CHIMERE.

The uncertainties in the observations $\mathbf{y}$ together with those in the observation operator $\mathcal{H}$, and the uncertainties in the prior estimate of the control vector $\mathbf{x}$ are assumed to have a Gaussian distribution Here, the first block of $\mathbf{B}$ corresponds to the logarithms of the anthropogenic flux factors. Each diagonal element is set at $(0.5)^2$: this variance value in the log-space corresponds to a factor ranging between 60%-164% in the emission space at the 1-day and model's grid scale. A second block is set for biogenic fluxes. On the diagonal, uncertainties are also set to a value of $(0.5)^2$, also corresponding to a factor ranging between 60%-164% in the emission space at the 1-day and model's grid scale. We account for spatial correlations in the uncertainties both for the anthropogenic and the natural parts. Spatial correlations are described by exponentially decaying functions with an e-folding length of 50 km over land and over the sea. Considering that part of the uncertainties in the



**Table 1.** Description of the inversions performed in this study.

| Name | Prior inventory | Satellite observations | Choice for the super-observation errors |
|------|-----------------|------------------------|------------------------------------------|
| OMI-A | TNO-GHGco-v3 for year 2005 | OMI-QA4ECV for year 2019 | conservative |
| OMI-B | TNO-GHGco-v3 for year 2005 | OMI-QA4ECV for year 2019 | optimistic |
| OMI-C | TNO-GHGco-v3 for January 2005 | OMI-QA4ECV for January 2005 | optimistic |
| OMI-D | TNO-GHGco-v3 for January 2019 | OMI-QA4ECV for January 2019 | optimistic |
| TROPOMI-A | TNO-GHGco-v3 for year 2005 | TROPOMI-RPRO-v2.4 for year 2019 | conservative |
| TROPOMI-B | TNO-GHGco-v3 for year 2005 | TROPOMI-RPRO-v2.4 for year 2019 | optimistic |

emission inventories are related to emission factors, and that uncertainties in such factors lead to strong spatial correlations over long distances for specific emission sectors, this value could seem to be too small. However, such spatial correlations

can be decreased when merging the different emission sectors, and isotropic correlations decreasing with distance may not be good representations of the spatial patterns of uncertainties in the highly heterogeneous maps of total anthropogenic emissions. Furthermore, using a relatively small value for the spatial correlation length is conservative: it limits, in the inversion process, the implicit spatial extrapolation and smoothing of the information from the satellite data for the correction of the prior emission estimate via the **B** matrix, which is challenging to characterize (Super et al., 2024), but it does not limit the corrections of the

emissions within the footprint of the satellite observations. Finally, in the third block of **B** for initial conditions, the variances are set at 20%.

The variance of the observation errors corresponding to individual super-observations in **R** is the quadratic sum of the error we have assigned to the OMI or TROPOMI super-observations, and of an estimate of the errors from the observation operator. We assume that the observation operator error is dominated by the chemistry-transport modeling error: it is set at 20% of

the retrieval value, as in Fortems-Cheiney et al. (2021b). The monthly averages of the observation errors in **R** are shown in Figure 5 for the different configurations associated with the errors assigned to the OMI or TROPOMI super-observations (see Section 2.4.3). It is interesting to note that the TROPOMI-A configuration, ignoring the number of observations to reduce the error associated with the super-observation, presents lower errors than the OMI-A one. This is explained by the better signal-to-noise ratio of TROPOMI compared to OMI (van Geffen et al., 2022b).

The different experiments performed in this study are presented in Table 1. The inversions are conducted using the variational mode of the CIF with the limited-memory quasi-Newton minimisation algorithm M1QN3 (Gilbert and Lemaréchal, 1989) for the minimization of the cost function $J$. At each iteration of this minimization, the CIF uses a CHIMERE simulation to compute $J$ for a new estimate of **x** and the adjoint code of CHIMERE to compute the gradient of $J$ for this new estimate of **x**. We impose a reduction of the norm of the gradient of $J$ by 90% as a constraint for the interruption of the minimization process but the

reduction of the norm of the gradient of $J$ actually often exceeds 95%.



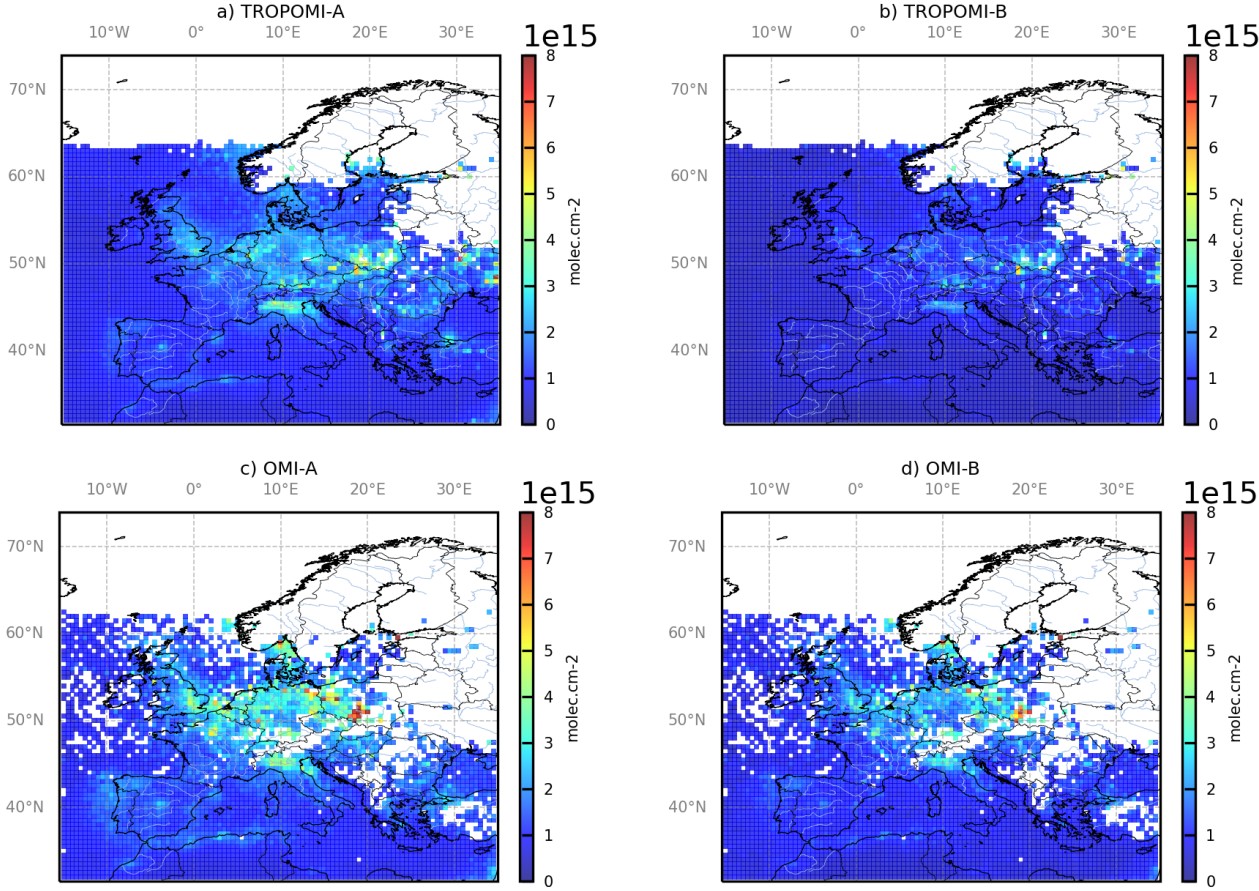

**Figure 5.** Monthly averages of the observation errors in **R** for the a) TROPOMI-A, b) TROPOMI-B, c) OMI-A and d) OMI-B configurations, in molec.cm$^{-2}$, for January 2019. The color scale is the same as for Figure 6. The observation errors in **R** include errors corresponding to the satellite retrievals and to the observation operator.

## 3 Results

### 3.1 Comparison between OMI and TROPOMI super-observations

The $NO_2$ TVCDS from OMI-QA4ECV and older versions of the TROPOMI products have been compared in the literature. Sekiya et al. (2022) reported lower concentrations in TROPOMI unofficial reprocessing product (version 1.2 beta) than in OMI-QA4ECV by 15% during April–May 2018 averaged over 60°S–60°N. As mentioned in the introduction, Lambert et al. (2021) reported $NO_2$ TVCDs from TROPOMI version 1.2 and 1.3 products systematically lower than from OMI-QA4ECV, with differences in terms of monthly $NO_2$ average reaching -40% over polluted regions in winter. Nevertheless, comparisons





of TROPOMI-v1.4 with OMI-QA4ECV observations show an improved consistency between the two retrievals. Over Europe, while the bias was of about -17% in January 2019 with version 1.3, it was of about -4% in January 2020 with version 1.4 (Lam-
bert et al., 2021). Here, we would like to characterize the consistencies between OMI-QA4ECV and TROPOMI-RPRO-v2.4, the more recent re-processing of the TROPOMI data.

OMI NO$_2$ and TROPOMI NO$_2$ super-observations present consistent geographical patterns (Fig. 6) with high concentrations over emission hot spots. However, the average magnitude of the NO$_2$ TVCDs is different. The NO$_2$ TVCDs in the OMI-QA4ECV product are often higher than in the TROPOMI-RPRO-v2.4 product (Fig. 6a vs Fig. 6d, Fig. 7), with a mean differ-
ence between the two products over the entire domain of about +15% in January 2019 (Fig. 6g). It could be explained by the fact that TROPOMI underestimates the NO$_2$ TVCDs over highly polluted areas. The last evaluation of the TROPOMI RPRO v2.4 product around the world still indicates significant biases of TROPOMI NO$_2$ TVCDs of typically +13% over clean areas to -40% over highly polluted areas (Lambert et al., 2023; van Geffen et al., 2022b), even if these bias estimates are reduced when MAX-DOAS profile data are vertically smoothed using the TROPOMI AKs.
Since the OMI and TROPOMI AKs are different, and since the spatio-temporal samplings of the two data sets are also differ- ent, we have chosen to conduct an indirect comparison of OMI and TROPOMI using the CHIMERE CTM as an intermediate. Interestingly, while the mean NO$_2$ TVCDs observed by OMI is higher by about +15% than the one observed by TROPOMI, the mean CHIMERE-OMI NO$_2$ TVCDs is lower by about 7% compared to the CHIMERE-TROPOMI ones over the entire domain in January 2019 (Fig. 6h). It can be explained by the fact that the AKs in TROPOMI-RPRO-v2.4 tend to be larger
above the first four levels than in OMI-QA4ECV (Figure 2).

## 3.2 Comparison between satellite super-observations and prior CHIMERE simulations

The CHIMERE-OMI and the CHIMERE-TROPOMI simulations, based on the TNO-GHGco-v3 inventory for the year 2005, both present higher NO$_2$ TVCDs than the OMI and TROPOMI super-observations for the year 2019 (Fig. 6). This is expected
since the NO$_x$ anthropogenic emissions have been decreased in Europe since 2005 (EEA, 2021). However, the magnitude of the discrepancies between the simulations and the satellite super-observations vary. For example, in January 2019 over the entire domain, the simulated CHIMERE-TROPOMI TVCDs are about 38% higher than TROPOMI super-observations while simulated CHIMERE-OMI TVCDs are about 23% higher than the OMI ones. Over the most polluted area, including parts of Western and Central Europe (0°E-20°E; 40°N-60°N, see the purple box in Fig. 6), the simulated CHIMERE-TROPOMI and
CHIMERE-OMI also present a strong positive relative difference of about +39% compared to TROPOMI and of about +29% compared to OMI respectively. TROPOMI therefore shows a stronger drop in the NO$_2$ TVCDs than OMI and consequently, the TROPOMI inversions would lead to lower NO$_x$ anthropogenic emissions than the OMI inversions.

The highest differences between the super-observations and the CHIMERE-OMI or CHIMERE-TROPOMI simulations are found in autumn and in winter, and particularly for the months of January, November and December (Fig. 7). It can be deduced
that the winter European NO$_2$ simulated TVCDs will be decreased both by the OMI and the TROPOMI inversions, even if the respective corrections to the prior emissions will be different in magnitude. However, in spring and in summer (e.g., for the





months of April, May, June, July and August), the monthly averages of the TROPOMI super-observations are always lower than the CHIMERE-TROPOMI NO$_2$ TVCDs while the monthly averages of the OMI super-observations remains close to the CHIMERE-OMI ones (Fig. 7b). The TROPOMI and OMI super-observations therefore will lead to different conclusions

concerning the potential reductions of NO$_x$ anthropogenic emissions during spring and summer since 2005.

### 3.3   Improvement of the fit between satellite super-observations and CHIMERE simulations

The inversions bring the simulated NO$_2$ TVCDs closer to OMI or to TROPOMI super-observations (Fig. 7). In general, the reduction of the bias between the super-observations and the CHIMERE simulations is higher in winter than in summer (Fig. 7).

In January 2019 over the entire domain, the mean bias between the TROPOMI-A super-observations and the CHIMERE-TROPOMI-A simulations is reduced by about 70%, while the mean bias is reduced by about 83% with the TROPOMI-B inversions. The mean bias between the OMI-A super-observations and the CHIMERE-OMI-A simulations is also reduced, by about 73%, while the mean bias is reduced by about 80% with the OMI B inversions. The TROPOMI-B and OMI-B corrections are higher than the TROPOMI-A and OMI-A ones, respectively. It can be explained by the lower error associated to the "B"

super-observations, giving more weight to the satellite data, compared to the "A" ones (see Section 2.4.3).

    To improve and optimize the fit between satellite super-observations and CHIMERE simulations the inversion system reduces the NOx anthropogenic emissions in the year 2019 (Fig. 8).




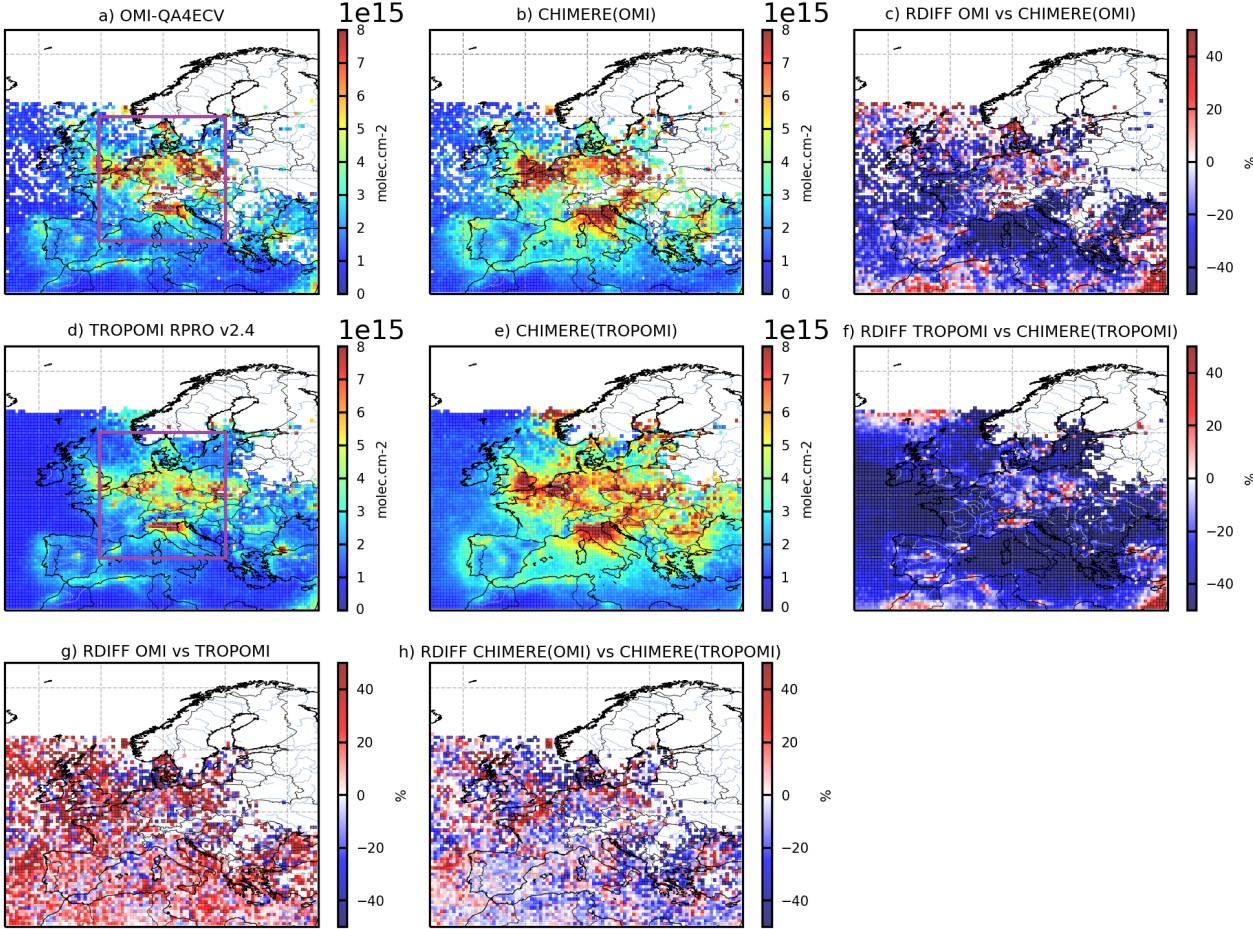

**Figure 6.** Monthly averages of NO$_2$ a) super-observations based on OMI and b) the CHIMERE-OMI simulation, where and when OMI-QA4ECV super-observations are available; d) super-observations based on TROPOMI RPROv2.4 and e) CHIMERE-TROPOMI simulations, where and when TROPOMI super-observations are available, in molec.cm$^{-2}$. Monthly averages for January 2019 of the relative differences between c) the super-observations from OMI-QA4ECV NO$_2$ TVCDs and the CHIMERE-OMI simulation, f) the super-observations from TROPOMI NO$_2$ TVCDs and the CHIMERE-TROPOMI simulation, g) the OMI and TROPOMI super-observations and h) the CHIMERE-OMI and the CHIMERE-TROPOMI simulations, in %. The prior TNO-GHGco-v3 anthropogenic emission for the year 2005 are used to simulate the NO$_2$ TVCDs. The purple box shows Western and Central Europe (0°E-20°E; 40°N-60°N).



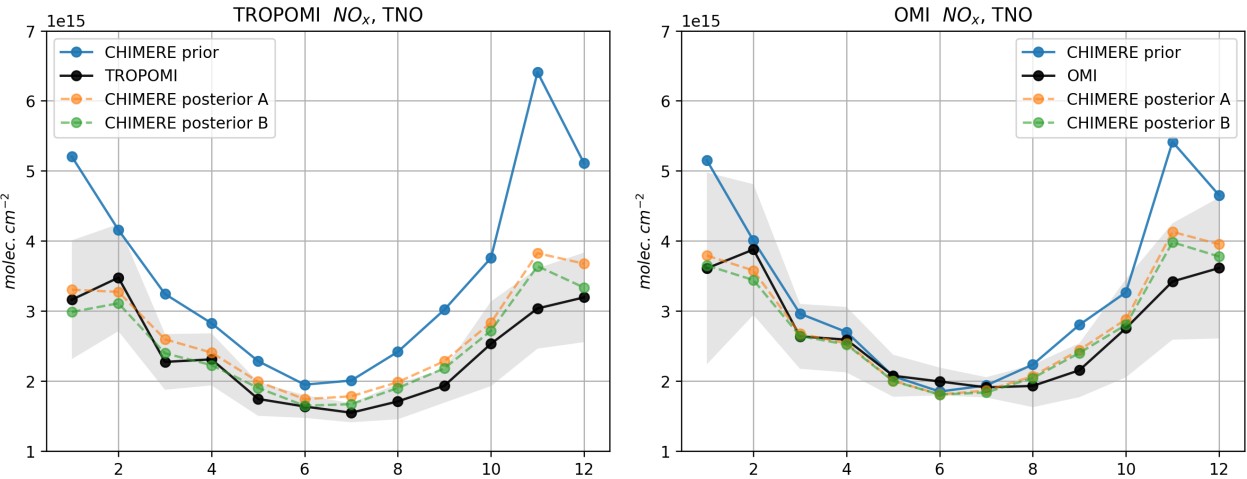

**Figure 7.** January to December 2019 times series of monthly averaged NO$_2$ TVCDs over Western and Central Europe (0°E-20°E; 40°N-60°N). Left: from TROPOMI RPROv2.4 super-observations (in black) and the CHIMERE-TROPOMI prior (in blue), CHIMERE-TROPOMI posterior A (in orange) and CHIMERE-TROPOMI posterior B (in green) simulations. Right: same as left but for OMI-QA4ECV super-observations, in molec.cm$^{-2}$. The prior simulations of TVCDs use TNO-GHGco-v3 anthropogenic emissions for the year 2005.





## 3.4 Posterior estimates of NO$_x$ European anthropogenic emissions in 2019

As the anthropogenic emissions contribute to about 95% of the total NO$_x$ emissions in Europe even in summer (see Sec-
tion 2.3), this section focuses on the inversion results in terms of comparisons between the prior anthropogenic NO$_x$ emissions
for 2005 and the posterior emissions from the OMI and TROPOMI inversions for the year 2019 in the European Union +
United Kingdom (EU-27+UK) area. Nevertheless, the inversion results do not seem to indicate missing sources in the biogenic
emissions, even if we do not include a specific component of NO$_x$ emissions from agricultural soils in our prior estimate of
NO$_x$ emissions (see Section 2.2). The annual budget for biogenic emissions are in fact only changed by a few percents both by
the OMI and TROPOMI inversions (not shown).

The posterior emissions from the OMI and TROPOMI inversions are compared to the emission estimates from the TNO-
GHGco-v3 inventory for the year 2019 (Tab. 3, Fig. 9). At the European scale (Fig. 9) and at the national scale (Tab. 3), both
the posterior anthropogenic NO$_x$ emissions from the OMI and TROPOMI inversions for the year 2019 are lower than the prior
ones for the year 2005.

When assimilating OMI-A super-observations, the NO$_x$ anthropogenic emissions for EU-27+UK and for Western Europe
are decreased by about 13% and 17%, respectively, between 2005 and 2019 (Table 3). Similar results are obtained when
assimilating OMI-B super-observations (decrease by about 16% and 21%, respectively, Table 3). These decreases of emissions
between 2005 and 2019 are higher than the decrease estimated by Miyazaki et al. (2017), finding a negative change of only
-0.1% in Europe (defined as 10°W–30°E, 35–60°N in their study) and -8.8% in Western Europe between 2005 and 2014.
However, the OMI-A and the OMI-B posterior NO$_x$ emissions over EU-27+UK respectively remain +53% and +47% higher
than the estimation of the TNO-GHGco-v3 inventory for 2019. To support the assumption that the positive bias between the
CHIMERE simulations driven by the inventory for 2005 and the satellite observations in 2019 (seen in Figure 6) is mainly
related to the decrease of the European emissions, we have performed two sensitivity tests for one month in January: the first,
called OMI-C, uses CHIMERE simulations driven by the TNO inventory for 2005 and OMI observations for 2005 (Table 1),
the second, called OMI-D, uses CHIMERE simulations driven by the TNO inventory for 2019 and OMI observations for 2019
(Table 1). In these cases, the NO$_x$ anthropogenic emissions for EU-27+UK in January 2005 and in January 2019 are both
decreased by about 6% by the inversions (Table 2). Using the same configuration of the **R** covariance matrix, these corrections
are much smaller than the correction of about -21% reached in the case OMI-B for the month of January 2019 (Table 2,
Figure 9), when using prior emissions from the TNO inventory for 2005 and OMI observations for 2019 (Table 1). This result
therefore shows that the positive bias between the CHIMERE simulations driven by the inventory for 2005 and the satellite
observations in 2019 is mainly due to the decline in European NO$_x$ emissions since 2005.

At the pixel scale both over urban and rural areas (Fig. 8) and at the national or European scales (Table 3, Fig. 9), the
decreases of NO$_x$ anthropogenic emissions estimated from the OMI inversions between 2005 and 2019 are lower than from
the TROPOMI ones. This can be explained by the fact that the relative differences between TROPOMI super-observations
and the CHIMERE-TROPOMI simulations are larger than between OMI super-observations and the CHIMERE-OMI ones
(see Section 3.2), due to different cloud pressures and albedos affecting the averaging kernels (see Section 2.4.3). As OMI



**Figure 8.** Monthly mean relative corrections to the prior NO$_x$ emissions for year 2005 from the a) TROPOMI-A and from the b) OMI-A inversions, calculated as (posterior-prior)/prior, in %, for top) the month of January 2019 and bottom) the entire year 2019.





**Table 2.** $NO_x$ anthropogenic emissions for EU-27+UK estimated from different sensitivity tests described in Table 1, in kteqNO$_2$, for the month of January 2005 or for the month of January 2019.

| Name of the inversions | Prior TNO-GHGco-v3 for January 2005 | Prior TNO-GHGco-v3 for January 2019 | Posterior using OMI-QA4ECV for January 2005 | Posterior using OMI-QA4ECV for January 2019 | Relative difference posterior-prior |
|---|---|---|---|---|---|
| TROPOMI-B | 1257 | - | - | 661 | -47% |
| OMI-B | 1257 | - | - | 993 | -21% |
| OMI-C | 1257 | - | 1180 | - | -6% |
| OMI-D | - | 736 | - | 696 | -6% |

super-observations are closer to the CHIMERE-OMI simulations, particularly in spring and summer, the OMI inversions make lower corrections in the $NO_x$ anthropogenic emissions than the TROPOMI ones. This can also be explained by the fact that i) TROPOMI presents a better coverage compared to OMI, with a much larger number of observations (Figure 4) and ii)
TROPOMI presents lower errors associated with its super-observations than OMI, even in the case where the error associated with the super-observation is not reduced depending on the number of observations (Figure 5).

Assimilating TROPOMI-A super-observations, the decrease of the $NO_x$ anthropogenic emissions for EU-27+UK and for Western Europe between 2005 and 2019 is indeed of about -32% and -42%, respectively. Assimilating TROPOMI-B super-observations even lead to higher decreases with -45% and -54%, respectively, for EU-27+UK and for Western Europe. It can
be explained by the lower error associated to the TROPOMI-B super-observations, giving more weight to the satellite data, compared to the TROPOMI-A ones (see Section 2.4.3). The decrease in $NO_x$ emissions from the TROPOMI-A inversions ranges from -9% for Cyprus to -54% for Belgium, while it ranges from -22% for Montenegro to -63% for Belgium from the TROPOMI-B ones. The TROPOMI-A posterior $NO_x$ emissions are closer to the TNO-GHGco-v3 inventory at the European scale compared to the OMI-A ones, but still with relative differences of about +19% for EU-27+UK. The TROPOMI-B poste-
rior emissions become consistent with the TNO-GHGco-v3 inventory, with a relative difference of only -4%.

Generally, the corrections provided by the inversions are stronger for Western and Southern countries than for Eastern or Northern ones (Table 3, Fig. 8). For example, the TROPOMI-B posterior emissions suggest higher annual budgets in several Central or Eastern European countries (e.g., Slovakia, Slovenia, Croatia) than those provided by TNO, based on official country emission reporting. It may be due to the fact that $NO_2$ TVCDs (Krotkov et al., 2016; Fortems-Cheiney et al., 2021a) and
the $NO_x$ emissions in these countries have not strongly decreased since 2005. It can also be explained by the cloud coverage limiting the number of satellite data over these countries (Figure 4). When the coverage of a country by OMI or TROPOMI super-observations is sparse, the posterior emissions indeed remain close to the prior emission estimates i.e. at their 2005 level. On the contrary, the TROPOMI-B inversions show lower annual budgets over Western European countries such as Belgium



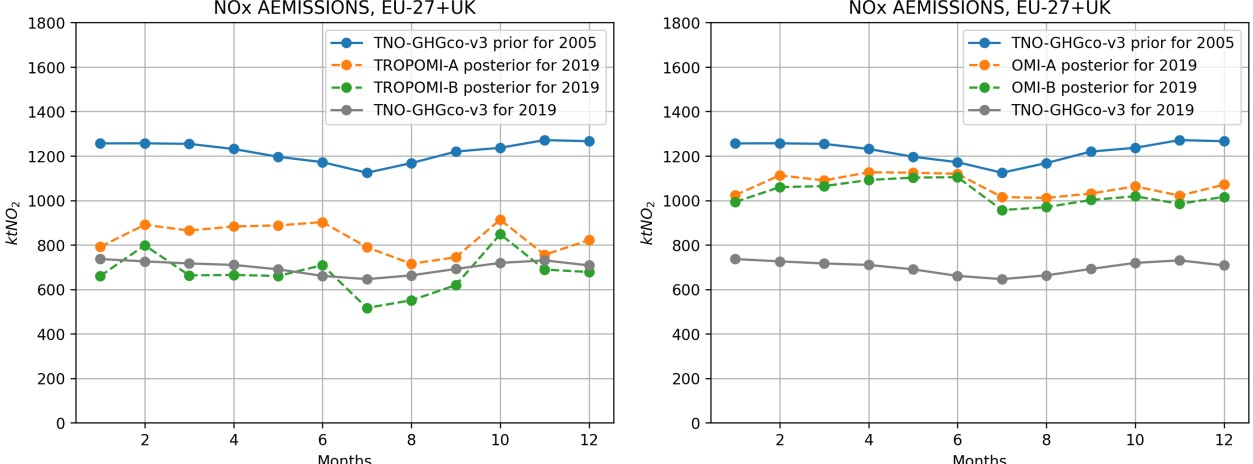

**Figure 9.** Left) January to December 2019 times series of monthly estimates of anthropogenic $NO_x$ prior emissions from the TNO-GHGco-v3 inventory for the year 2005 (in blue), posterior anthropogenic $NO_x$ emission estimates from the TROPOMI-A (in orange) and TROPOMI-B (in green) inversions, in kteqNO$_2$, for the EU-27+UK area. Right) same as left, but for posterior anthropogenic $NO_x$ emission estimates from the OMI-A (in orange) and OMI-B (in green) inversions.

and the Netherlands than TNO (Table 3), suggesting that $NO_x$ emissions could have been more reduced than officially reported in these countries.

Table 3: National anthropogenic prior $NO_x$ emission estimates from the TNO-GHGco-v3 inventory for the year 2005 and relative increments provided by the different inversions in %, for the year 2019. The relative differences between emission estimates of the TNO-GHGco-v3 anthropogenic for the year 2019 against year 2005 are given for information.

| Country | Prior TNO-GHGco-v3 2005 | Increments from the TROPOMI-A inversion | Increments from the TROPOMI-B inversion | Increments from the OMI-A inversion | Increments from the OMI-B inversion | Rdiff TNO 2019 vs 2005 |
|---|---|---|---|---|---|---|
| | kteqNO$_2$ | % | % | % | % | % |
| Albania | 32 | -19 | -36 | -7 | -10 | 4 |
| Austria | 224 | -30 | -43 | -12 | -15 | -42 |
| Belgium | 359 | -54 | -63 | -27 | -31 | -47 |
| Bulgaria | 175 | -15 | -29 | -5 | -7 | -47 |
| Bosnia-Herzegovina | 53 | -13 | -24 | -4 | -6 | -27 |
| Croatia | 116 | -19 | -32 | -6 | -9 | -40 |

…/…



| Country | Prior TNO-GHGco-v3 2005 | Increments from the TROPOMI-A inversion | Increments from the TROPOMI-B inversion | Increments from the OMI-A inversion | Increments from the OMI-B inversion | Rdiff TNO 2019 vs 2005 |
|---|---|---|---|---|---|---|
| | kteqNO$_2$ | % | % | % | % | % |
| Switzerland | 88 | -30 | -41 | -13 | -16 | -35 |
| Cyprus | 52 | -9 | -22 | -4 | -6 | -33 |
| Czech Republic | 271 | -27 | -37 | -9 | -11 | -45 |
| Germany | 1704 | -34 | -44 | -13 | -15 | -32 |
| Denmark | 346 | -37 | -49 | -11 | -13 | -47 |
| Spain | 1670 | -29 | -46 | -12 | -16 | -49 |
| Estonia | 82 | -20 | -30 | -4 | -5 | -32 |
| Finland | 239 | -15 | -24 | -3 | -3 | -42 |
| France | 1747 | -34 | -48 | -12 | -16 | -48 |
| United Kingdom | 2049 | -48 | -59 | -20 | -23 | -50 |
| Greece | 834 | -27 | -45 | -12 | -17 | -43 |
| Hungary | 161 | -22 | -34 | -7 | -10 | -38 |
| Ireland | 171 | -27 | -41 | -5 | -7 | -46 |
| Italy | 1575 | -26 | -41 | -12 | -15 | -47 |
| Lithuania | 59 | -14 | -23 | -2 | -3 | -25 |
| Latvia | 55 | -13 | -23 | -2 | -3 | -30 |
| Montenegro | 11 | -11 | -22 | -3 | -5 | -22 |
| Netherlands | 583 | -48 | -55 | -23 | -25 | -40 |
| Norway | 308 | -24 | -36 | -4 | -5 | -32 |
| Poland | 811 | -24 | -33 | -6 | -8 | -19 |
| Portugal | 515 | -23 | -42 | -8 | -12 | -43 |
| Romania | 314 | -15 | -27 | -4 | -6 | -31 |
| Serbia | 170 | -22 | -36 | -8 | -11 | -39 |
| Slovakia | 102 | -21 | -31 | -7 | -9 | -43 |
| Slovenia | 60 | -24 | -38 | -9 | -12 | -47 |
| Sweden | 315 | -24 | -35 | -6 | -7 | -36 |
| Turkey | 660 | -13 | -25 | -5 | -7 | -16 |
| Ukraine | 511 | -14 | -25 | -4 | -5 | -33 |
| Benelux | 955 | -50 | -58 | -24 | -27 | -43 |
| Western Europe | 4922 | -42 | -54 | -17 | -21 | -48 |



| Country | Prior TNO-GHGco-v3 2005 | Increments from the TROPOMI-A inversion | Increments from the TROPOMI-B inversion | Increments from the OMI-A inversion | Increments from the OMI-B inversion | Rdiff TNO 2019 vs 2005 |
|---|---|---|---|---|---|---|
| | kteqNO$_2$ | % | % | % | % | % |
| Central Europe | 3361 | -30 | -40 | -11 | -13 | -32 |
| Northern Europe | 1403 | -25 | -36 | -6 | -7 | -38 |
| Southern Europe | 5365 | -25 | -42 | -11 | -14 | -46 |
| Eastern Europe | 1238 | -19 | -28 | -5 | -6 | -32 |
| EU-27+UK | 14655 | -32 | -45 | -13 | -16 | -43 |



## 4 Conclusion

There are great expectations about the detection and the quantification of $NO_x$ emissions using $NO_2$ TVCDs from satellite observations and inverse systems. This study assesses the potential of the OMI-QA4ECV and TROPOMI satellite observations to improve the knowledge on European $NO_x$ emissions at the regional scale and to inform about the spatio-temporal variability

of $NO_x$ anthropogenic emissions from 2005 to 2019, at the resolution of 0.5° over Europe. Starting from European emission estimates from the TNO-GHGco-v3 inventory for the year 2005, regional inversions using the CIF coupled to CHIMERE CTM and assimilating satellite $NO_2$ TVCDs from OMI and TROPOMI have been performed to estimate the European annual and seasonal budgets for the year 2019.

Both the OMI and TROPOMI inversions show decreases in European $NO_x$ anthropogenic emission budgets between 2005

and 2019. Nevertheless, the magnitude of the reductions of the $NO_x$ anthropogenic emissions are different with OMI and TROPOMI data, with decreases in EU-27+UK between 2005 and 2019 of 16% and 45% respectively. The decrease of $NO_x$ anthropogenic emissions estimated from the TROPOMI inversions are substantially higher than from the OMI inversions. This is explained by i) the fact that the differences betzeen CHIMERE-TROPOMI simulations and the TROPOMI super-observations are larger than between CHIMERE-OMI and OMI super-observations, due to different AKs, ii) the better coverage

of TROPOMI compared to OMI and iii) the lower errors associated with the TROPOMI super-observations compared to OMI.

The TROPOMI-B inversions, giving weight to the satellite data, become consistent with the independent TNO-GHGco-v3 inventory for the year 2019, with annual budgets for EU-27+UK showing absolute relative difference of only 4%. These TROPOMI inversions are therefore in agreement with the magnitude of the decline in $NO_x$ emissions declared by countries, when aggregated at the European scale.

Our results —with OMI and TROPOMI data leading to different magnitudes of corrections on $NO_x$ anthropogenic emissions —suggest that more observational constraints would be required to sharpen the European emission estimates. Observational information from future satellite missions such as Sentinel-4 on board geostationary satellites would increase the number of observations for better constraining the $NO_x$ emissions in particular for Eastern and Northern countries.

## 5 Author contributions

AFC and GB conceptualized the study and carried out the results analysis. AFC carried out the inversions. EP, RP, AB, IP and AM developed the CIF inversion system, including preprocessing for fluxes and satellite observations. HDvdG and SD provided the TNO-GHGco-v3 inventory used as prior emissions in this study. All the co-authors contributed to writing the manuscript.

## 6 Data availability

OMI-QA4ECV data are freely available through the website www.qa4ecv.eu and http://temis.nl/qa4ecv/no2.html, (Boersma et al., 2017). TROPOMI-RPRO-v2.4 data are freely available through the website https://identity.dataspace.copernicus.eu. The





TNO-GHGco-v3 inventory (Super et al., 2020) is available upon request from TNO (contact: Hugo Denier van der Gon, hugo.deniervandergon@tno.nl).

## 7  Code availability

The CHIMERE code is available here: http://www.lmd.polytechnique.fr/chimere/ (Menut et al., 2013; Mailler et al., 2017). The CIF inversion system (Berchet et al., 2021) is available at https://doi.org/10.5281/zenodo.6304912 (Berchet et al., 2022).

## 8  Financial support

A large part of the development and analysis were conducted in the frame of the H2020 VERIFY and COCO2 projects, funded by the European Commission Horizon 2020 research and innovation programme, respectively under agreement number 776810
and 958927, and in the frame of the World Emission project funded by the European Space Agency. This study has received funding from the French ANR project ARGONAUT under grant agreement No ANR-19-CE01-0007 and from the French PRIMEQUAL project LOCKAIR under grant agreement No 2162D0010. This work was also supported by the CNES (Centre National d'Etudes Spatiales), in the frame of the TOSCA ARGOS project.

## 9  Acknowledgements

We acknowledge the OMI-QA4ECV and the TROPOMI group for the production of the $NO_2$ retrievals. We wish to thank all the persons involved in the preparation, coordination and management of the H2020 VERIFY and COCO2 projects, as well as the ESA World Emission project. This work was granted access to the HPC resources of TGCC under the allocations A0140102201 made by GENCI. Finally, we wish to thank Julien Bruna (LSCE) and his team for computer support.

## 10  Competing Interests

The authors declare that they have no conflict of interest.



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
