# Peer review of "Decrease of the European $NO_x$ anthropogenic emissions between 2005 and 2019 as seen from the OMI and TROPOMI $NO_2$ satellite observations"

_EGUsphere, 2024_

## Author Comment (AC1)

**Review #1**

**We wish to thank the referee for his/her helpful comments. His/her full comments are copied hereafter in normal black font, and our responses are inserted in between in blue and bold font.**

Overall, it is clear that a large amount of work was done by the authors. However, in its current state, I cannot recommend publication. I do encourage the authors to carefully consider my comments and resubmit the revised manuscript.

A major concern of mine is that this article doesn't present new science, so it may be better suited to another journal, such as Earth System Science Data (ESSD). The title and introduction actually sound somewhat appropriate for ACP, but the manuscript content is really not appropriate for ACP.

**We feel that this comment, and also some of the comments below, are driven by the fact that the title of the manuscript was misleading. The main part of the analysis assess the potential of the satellite data to support the estimate of the decrease of the $NO_x$ emissions over time, rather than to the analysis of this decrease itself. The title has been revised accordingly, and the introduction and conclusion have also been improved to make it clearer.**

Another issue is that the manuscript is poorly organized and, therefore, difficult to read. ACP gives handy recommendations on the structure of a manuscript: https://www.atmospheric-chemistry-and-physics.net/policies/guidelines_for_authors.html. I highly suggest that the authors review these recommendations. Here are two examples of what I mean:

- The title doesn't reflect what the paper is about. The paper title indicates that the paper is focused on discussing the causes of trends in European NOx emissions. At least, that's what I thought when reading the title. This isn't the case. Please revise the title to reflect the actual manuscript content.

**We agree that the title of the article was potentially misleading and we have changed it. It is now: "Assessing the ability to quantify the decrease of $NO_x$ anthropogenic emissions in 2019 compared to 2005 using OMI and TROPOMI satellite observations".**

- The paper is wordy (by 30-50%); the introduction is particularly wordy and needs paragraph indents for clarity. Note that a goal of technical writing is to clearly and concisely convey a particular message. As an example, the purpose of your paper should be given in one sentence, such as in the abstract and as a topic sentence of a paragraph in the introduction. I had to piece it together over a very long paragraph, but I still don't know the overall goal of the paper given the vague title, vague abstract, and vague conclusions. An overview diagram of the steps in your work would be helpful.

**We had made strong efforts to build an introduction providing a comprehensive overview of the context for the study: the $NO_2$ concentration and $NO_x$ emission monitoring, the reduction of $NO_x$ emissions, the state-of-the-art methods for the quantification of the $NO_x$ emissions and the main limitations of both bottom-up inventory and of atmospheric inversions using satellite observations. We nevertheless understand the reviewer's comments and we have shortened and reformulated the introduction. The technical paragraph about**

the similarities and the differences between the OMI and the TROPOMI algorithms and datasets has been moved to Section 2.

The objectives of the study have been reformulated and a diagram providing an overview of the inversion and experimental frameworks, and of the results has been inserted as the new Figure 1. We have also indented paragraphs in the introduction for clarity.

[Figure]

**Figure 1. Simplified scheme of the iterative minimization in the CIF-CHIMERE inversion system and illustration of the decrease of the European NO$_x$ emissions estimated both by TNO and by the OMI and TROPOMI-based inversions. The different inversions are described in Table 1. The numbers express the anthropogenic NO$_x$ emission estimates for the EU-27+UK area in kteqNO$_2$. The numbers in brackets express the difference of the anthropogenic NOx emission estimates for the EU-27+UK in 2019 compared to 2005, in %.**

Another major concern is the lack of validation of the data products and emission estimates. Why don't you compare results with independent in situ observations, such as from the Pandora network? That would certainly strengthen your conclusions about how your method impacts the emissions estimates. In fact, you say in the last sentence of the abstract "…our results —with OMI and TROPOMI data leading to different magnitudes of corrections on NOx anthropogenic emissions—suggest that more observational constraints would be required to sharpen the European emission estimates." You haven't even used the existing observational constraints !

The comparison between our type of regional CTM (here at 0.5° resolution) and typical in situ air quality observations in view to evaluate the emission estimates, and consequently the

validation of our inversions using such data or the assimilation of such data can be complex due, e.g., to the difficult representation of these observation close to the surface and to source in the model. As described by Lamsal et al. (2008), the instrument most commonly used for routine measurements of $NO_2$ is a chemiluminescence analyzer, a measurement technique which exhibits significant interference from other reactive oxidized nitrogen-containing species ($NO_z$) such as peroxyacetyl nitrate (PAN) and nitric acid ($HNO_3$). It could consequently result in an overestimation of ambient $NO_2$ concentrations (Savas et al. 2023). The joint assimilation of surface and satellite observation would raise further theoretical and technical challenges. Specific studies are needed to properly handle such a use of the in situ observations, which explains why we have a long list of publications focusing on the use of the satellite data (see our introduction, van der A et al., 2008; Lamsal et al., 2011; Mijling et al., 2012; Mijling et al., 2013; Miyazaki et al., 2017; Plauchu et al., 2024; van der A et al., 2024).

Furthermore, here, the analysis are focused on the assessment of the ability to provide information on the decrease of $NO_x$ anthropogenic emissions in 2019 compared to 2005 based on satellite $NO_2$ observation (that from the OMI and TROPOMI instruments). Given their spatial coverage and spatial resolution, and their vertical representativity, these satellite observations— are likely more suitable for inversions at the targeted scales than the surface measurements in the area of analysis. This rationale is now better expressed with our new title and in our revision of the introduction.

Lamsal, L.N.; Martin, R.V.; van Donkelaar, A.; Steinbacher, M.; Celarier, E.A.; Bucsela, E.; Dunlea, E.J.; Pinto, J.P. Ground-level nitrogen dioxide concentrations inferred from the satellite-borne Ozone Monitoring Instrument. J. Geophys. Res. 2008, 113, D16308.

Section 2.1: What are the strengths and limitations of this inversion system for your work? Has it been applied and validated with independent observations (e.g., Pandora)? Why are you using it relative to other inversion systems?

As now stated in the introduction, our inversion system combines both the advantages:

- of solving large dimensional inversion problems, i.e. controlling emissions at relatively high temporal and spatial resolution and assimilating a large amount of observations, via the variational mode of the CIF
- of simulating $NO_2$ concentrations and the sensitivities of $NO_2$ tropospheric columns to surface emissions at a relatively high spatial resolution with a chemistry scheme, based on the Eulerian regional CTM CHIMERE, and its adjoint code.

Our inversion system is therefore well designed to estimate $NO_x$ emissions at the 0.5° spatial resolution and assimilate satellite $NO_2$ retrievals, taking into account the non-linearities of the $NO_x$ chemistry. The CIF-CHIMERE variational inversion configuration has already been used for the estimation of the emissions of $NO_x$ (Savas et al., 2023; Plauchu et al., 2024) but also of other species such as CO (Fortems-Cheiney et al., 2023) and $CO_2$ (MacGrath et al., 2023). These information have been added in the introduction.

Our study assesses the potential of the assimilation of OMI and TROPOMI satellite observations to provide information on the decrease of $NO_x$ anthropogenic emissions in 2019

compared to 2005: we evaluate our emissions estimates for the year 2019 through comparisons to the independent estimates from the TNO inventory for this same year, at the European and at the national scales.

The title of Section 3.4 has been changed and is now: "Posterior estimates of NOx European anthropogenic emissions in 2019: evaluation with comparisons to the TNO-GHGco-v3 inventory"

Fortems-Cheiney, A., Broquet, G., Potier, E., Plauchu, R., Berchet, A., Pison, I., Denier van der Gon, H., and Dellaert, S.: CO anthropogenic emissions in Europe from 2011 to 2021: insights from Measurement of Pollution in the Troposphere (MOPITT) satellite data, Atmos. Chem. Phys., 24, 4635–4649, https://doi.org/10.5194/acp-24-4635-2024, 2024.

Plauchu, R., Fortems-Cheiney, A., Broquet, G., Pison, I., Berchet, A., Potier, E., Dufour, G., Coman, A., Savas, D., Siour, G., and Eskes, H.: NOx emissions in France in 2019–2021 as estimated by the high spatial resolution assimilation of TROPOMI NO2 observations, EGUsphere [preprint], https://doi.org/10.5194/egusphere-2024-103, 2024.

McGrath, M. J., Petrescu, A. M. R., Peylin, P., Andrew, R. M., Matthews, B., Dentener, F., Balkovič, J., Bastrikov, V., Becker, M., Broquet, G., Ciais, P., Fortems-Cheiney, A., Ganzenmüller, R., Grassi, G., Harris, I., Jones, M., Knauer, J., Kuhnert, M., Monteil, G., Munassar, S., Palmer, P. I., Peters, G. P., Qiu, C., Schelhaas, M.-J., Tarasova, O., Vizzarri, M., Winkler, K., Balsamo, G., Berchet, A., Briggs, P., Brockmann, P., Chevallier, F., Conchedda, G., Crippa, M., Dellaert, S. N. C., Denier van der Gon, H. A. C., Filipek, S., Friedlingstein, P., Fuchs, R., Gauss, M., Gerbig, C., Guizzardi, D., Günther, D., Houghton, R. A., Janssens-Maenhout, G., Lauerwald, R., Lerink, B., Luijkx, I. T., Moulas, G., Muntean, M., Nabuurs, G.-J., Paquirissamy, A., Perugini, L., Peters, W., Pilli, R., Pongratz, J., Regnier, P., Scholze, M., Serengil, Y., Smith, P., Solazzo, E., Thompson, R. L., Tubiello, F. N., Vesala, T., and Walther, S.: The consolidated European synthesis of CO2 emissions and removals for the European Union and United Kingdom: 1990–2020, Earth Syst. Sci. Data, 15, 4295–4370, https://doi.org/10.5194/essd-15-4295-2023, 2023.

Savas, D., Dufour, G., Coman, A., Siour, G., Fortems-Cheiney, A., Broquet, G., Pison, I., Berchet, A., & Bessagnet, B. (2023). Anthropogenic NOx Emission Estimations over East China for 2015 and 2019 Using OMI Satellite Observations and the New Inverse Modeling System CIF-CHIMERE. Atmosphere, 14(1), 154. https://doi.org/10.3390/atmos14010154

---

## Author Comment (AC2)

**Review #2**

We wish to thank the referee for his/her helpful comments. His/her full comments are copied hereafter in normal black font, and our responses are inserted in between in blue and bold font.

This manuscript presents $NO_x$ emission inversion experiments driven by two different satellite $NO_2$ column observational data sets. By performing inversions for 2019 using prior emissions for 2005, this study addresses the question to what extent bottom-up reported $NO_x$ emission reductions can be reconciled using satellite $NO_2$ observations in an emission inversion. This study is an important contribution to the literature and is well within the scope of ACP, with the community increasingly focusing on reconciling top-down and bottom-up emission estimates. The methodology is mostly scientifically sound, but the presentation of the methodology and results in the manuscript is unclear. This paper can be considered for publication in ACP after a revision focused primarily on restructuring the manuscript and improving the clarity of writing. Please find below the specific issues to be addressed in a revised manuscript.

**We thank the referee for his/her positive general comments.**

**Methodological issues:**

- While I agree with the authors that soil emissions are a relatively minor term on annual time scales, this is not the case for the spring and summer months when biogenic and agricultural emissions peak and anthropogenic emissions are lowest. Previous studies (e.g., Silvern et al. 2019) found that accounting for soil emissions is important to reconcile simulated and observed $NO_2$ column trends. Please discuss the effect of omitting agricultural soil emissions on the estimated $NO_x$ emission reduction trends.

**We agree with the reviewer about the importance of accounting for soil emissions to reconcile simulated and observed $NO_2$ column trends, as shown by Silvern et al. (2019). This is in agreement with our study, published in 2021, confronting trends in OMI $NO_2$ tropospheric columns over the 10-year 2008–2017 period in Europe to those of tropospheric columns simulated with the regional chemistry-transport model CHIMERE and state-of-the-art emission estimates for anthropogenic and biogenic $NO_x$ emissions (Fortems-Cheiney et al., 2021).**

**Actually, we do take soil emissions from MEGAN into account, as explained in Section 2.2. The impact of agriculture on soil emissions is neither provided by the TNO inventory nor by MEGAN. Therefore we do not include a specific agricultural soil $NO_x$ emissions component in our prior estimation of the $NO_x$ soil emissions. However, the atmospheric inversions controls these soil emissions (separately from the anthropogenic emissions) and could have increased them if it had supported the decrease of the misfits to the satellite observations above agricultural areas, which was not the case. As stated in Section 3.4,**

« the annual budget for biogenic emissions are only changed by a few percent both by the OMI and TROPOMI inversions. »,

In this study, the analysis are focused on the quantification of the decrease of $NO_x$ European emissions in 2019 compared to 2005. As the anthropogenic emissions strongly contribute to the total $NO_x$ emissions in Europe even in summer, we assume that the differences between the 2005 and the 2019 budgets are mainly due to anthropogenic emissions and not to the biogenic ones. Finally, the level of distinction between the anthropogenic and biogenic emissions in the inversions is one of the sources of uncertainty in the estimate of the anthropogenic emissions: as such, it is indirectly assessed together with other sources of uncertainty in the evaluation with comparisons to the TNO inventory.

We have added these information in the conclusion: "As the anthropogenic emissions strongly contribute to the total $NO_x$ emissions in Europe even in summer, we assume that the differences between the 2005 and the 2019 budgets are mainly due to anthropogenic emissions and not to the biogenic ones. However, the level of distinction between the anthropogenic and biogenic emissions in the inversions is one of the sources of uncertainty in the estimate of the anthropogenic emissions. The corrections provided by the inversions to the prior emissions can also be limited by the cloud coverage affecting the OMI or TROPOMI observations, by errors in the OMI or TROPOMI data and by errors in the CTM. Finally, the set-up of error covariance matrices could also have a strong impact on the emission estimates resulting from the inversions."

- In my view, this study is missing a section on the performance of the CIF-CHIMERE system using the two assimilated datasets. Please include this, e.g. by presenting error reductions or a comparison with independent (surface) observations.

The derivation of the statistics of uncertainty in the emission estimates from the inversion ("the posterior uncertainty"), and thus of the "error reduction" (the relative difference between prior and posterior uncertainties) is highly challenging when solving high dimensional inversion problems in variational mode (Rayner et al. 2019). In principle, the posterior uncertainty can be derived approximately using a Monte Carlo approach (Chevallier et al. 2007). However, because of the large computational cost of such a Monte Carlo framework, we have chosen not to perform it.

The comparison between our type of regional CTM (here at 0.5° resolution) and typical in situ air quality observations in view to evaluate the emission estimates, and consequently the validation of our inversions using such data or the assimilation of such data can be complex due for example to the difficult representation of these observation close to the surface and to source in the model. In addition, as described by Lamsal et al. (2008), the instrument most commonly used for routine measurements of $NO_2$ is a chemiluminescence analyzer, a measurement technique which exhibits significant interference from other reactive oxidized nitrogen-containing species ($NO_z$) such as peroxyacetyl nitrate (PAN) and nitric acid ($HNO_3$). It could consequently result in an overestimation of ambient $NO_2$ concentrations (Savas et al. 2023). Specific studies are

needed to properly handle such a use of the in situ observations, which explains why we have a long list of publications focusing on the use of the satellite data (see our introduction, van der A et al., 2008; Lamsal et al., 2011; Mijling et al., 2012; Mijling et al., 2013; Miyazaki et al., 2017; Plauchu et al., 2024; van der A et al., 2024). .

Furthermore, here, the analysis are focused on the assessment of the ability to provide information on the decrease of NOₓ anthropogenic emissions in 2019 compared to 2005 based on satellite NO₂ observation (that from the OMI and TROPOMI instruments). Given their spatial coverage and spatial resolution, and their vertical representativity, these satellite observations— are likely more suitable for inversions at the targeted scales than the surface measurements in the area of analysis. This rationale is now better expressed with our new title and in our revision of the introduction.

Chevallier, F., F.-M. Bréon, and P. J. Rayner (2007), Contribution of the Orbiting Carbon Observatory to the estimation of CO₂ sources and sinks: Theoretical study in a variational data assimilation framework, *J. Geophys. Res.*, 112, D09307, doi:10.1029/2006JD007375.

Lamsal, L.N.; Martin, R.V.; van Donkelaar, A.; Steinbacher, M.; Celarier, E.A.; Bucsela, E.; Dunlea, E.J.; Pinto, J.P. Ground-level nitrogen dioxide concentrations inferred from the satellite-borne Ozone Monitoring Instrument. J. Geophys. Res. 2008, 113, D16308.

Rayner, P. J., Michalak, A. M., and Chevallier, F.: Fundamentals of data assimilation applied to biogeochemistry, Atmos. Chem. Phys., 19, 13911–13932, https://doi.org/10.5194/acp-19-13911-2019, 2019.

**Writing issues:**

- The title does not reflect the contents of the paper: TROPOMI observations were not available in 2005, and the paper does not focus on detecting NOₓ emission decreases but rather on reconciling top-down- and bottom-up-derived emission trends. Please revise.

We agree that the title of the article was potentially misleading and we have changed it. It is now: "Assessing the ability to quantify the decrease of NOₓ anthropogenic emissions in 2019 compared to 2005 using OMI and TROPOMI satellite observations".

- Methodology: there is little coherence between these subsections.

Explanations have been added at the beginning of Section 2 to show the coherence between the subsections: "The principle of our traditional atmospheric inversion approach is to correct « a priori » estimates of the emission maps, also denoted "prior emissions", to reduce differences between atmospheric observations and their simulations with a chemistry-transport models (CTMs) fed with the estimates of emission maps. It relies on a specific Bayesian inversion algorithm, on satellite observations of the atmospheric densities of NO2, and on a regional CTM, as shown in Figure 1. We detail these different components of our inversion framework in Section 2: the CIF-CHIMERE variational

inversion system, the prior estimates of the $NO_x$ emissions in Europe for the year 2005, the configuration of the CHIMERE CTM and of its adjoint code for the simulation of $NO_2$ concentrations and of their sensitivity to the emissions estimates, and the OMI-QA4ECV and TROPOMI satellite observations for the year 2019. The process ensuring a suitable comparisons between the simulations and the satellite observations -the aggregation of the observations into super-observations and the application of the averaging kernels from the satellite retrievals to the vertical columns of the model simulations - are also explained. Finally, we provide details about our configuration for the inversion of $NO_x$ emissions over Europe."

The title of Section 2.4.3 has also been changed for clarity, it is now « Choices made to ensure a consistent comparison between simulated and observed $NO_2$ TVCD » and no more « Comparison between simulated and observed $NO_2$ TVCDs ».

- Please consider giving more descriptive names to your inversion experiments and including a conceptual diagram summarizing your methods.

As recommended, the names of the experiments have been changed in the text and in the figures. We now use the terms "optimistic" or "conservative", respectively when the super-observation uncertainty is reduced compared to that of individual observations or not. Due to these new names, few sentences have been slightly changed in Section 3.3 to simplify the reading.

We have also included a diagram providing an overview of the inversion and experimental frameworks, and of the results in the new Figure 1.

[Figure]

**Figure 1. Simplified scheme of the iterative minimization in the CIF-CHIMERE inversion system and illustration of the decrease of the European $NO_x$ emissions estimated both by TNO and by the OMI and TROPOMI-based inversions. The different inversions are described in Table 1. The numbers express the anthropogenic $NO_x$ emission estimates for the EU-27+UK area in kteqNO₂. The numbers in brackets express the difference of the anthropogenic NOx emission estimates for the EU-27+UK in 2019 compared to 2005, in %.**

**Minor issues:**

Introduction: please introduce non-anthropogenic $NO_x$ emissions before the first use of biogenic emissions in line 55.

**This has been done.**

Section 2.1: please add more detail on the CIF-CHIMERE setup. Since $NO_x$ chemistry is non-linear, how do $NO_x$ emission changes impact $NO_2$ columns via the chemistry scheme?

**Our inversion system provides the advantages of simulating NO₂ concentrations and the sensitivities of NO₂ tropospheric columns to surface emissions at a relatively high spatial resolution with a chemistry scheme, based on the Eulerian regional CTM CHIMERE and the MELCHIOR-2 chemical scheme, and their adjoint codes.**

**The MELCHIOR-2 chemical scheme used here indeed accounts for non-linear relationships between NO$_x$ (NO and/or NO$_2$) emission changes and NO$_2$ TVCDs via reactions with hydroxyl (OH) radicals but also with other direct or indirect NO$_x$ sinks associated with other species, such as ozone (O$_3$) or the HO$_2$ radical. These information have been added in Section 2.3.**

L217-218: what is the goal of reprocessing the NO$_2$ column observations using meteorological data?

**These goals are explained in the user's guides of OMI and TROPOMI retrievals. For example, according to Lambert et al. (2024), the TROPOMI NO$_2$ retrieval explicitly accounts for temperature effects by using the co-located temperature profiles in troposphere and stratosphere from the ECMWF meteorological analyses in the DOAS retrieval of the slant column density.**

**Lambert, J.-C., A. Keppens, S. Compernolle, K.-U. Eichmann, M. de Graaf, D. Hubert, B. Langerock, A. Ludewig, M.K. Sha, T. Verhoelst, T. Wagner, C. Ahn, A. Argyrouli, D. Balis, K.L. Chan, M. Coldewey-Egbers, I. De Smedt, H. Eskes, A.M. Fjæraa, K. Garane, J.F. Gleason, F. Goutail, J. Granville, P. Hedelt, K.-P. Heue, G. Jaross, Q. Kleipool, ML. Koukouli, R. Lutz, M.C Martinez Velarte, K. Michailidis, A. Pseftogkas, S. Nanda, S. Niemeijer, A. Pazmiño, G. Pinardi, A. Richter, N. Rozemeijer, M. Sneep, D. Stein Zweers, N. Theys, G. Tilstra, O. Torres, P. Valks, J. van Geffen, C. Vigouroux, P. Wang, and M. Weber : Quarterly Validation Report of the Copernicus Sentinel-5 Precursor Operational Data Products #24: April 2018 – August 2024. S5P MPC Routine Operations Consolidated Validation Report series, Issue #24, Version 24.00.00, 212 pp., 16 September 2024**

L350-351: wouldn't OMI also underestimate columns in polluted areas, due to the coarse-resolution a priori NO$_2$ profiles? Why is this problem larger for TROPOMI?

**Indeed, it is probable that OMI also underestimates NO$_2$ columns in polluted areas, as OMI and TROPOMI both use the same coarse-resolution a priori NO$_2$ profiles. Nevertheless, these coarse-resolution a priori NO$_2$ profiles can have a stronger impact on the TROPOMI columns than on the OMI ones, as TROPOMI has a finer spatial resolution than OMI.**

**OMI vs TROPOMI evaluations have shown that TROPOMI provides smaller NO$_2$ TVCDs compared to OMI over polluted areas, at least with previous TROPOMI versions (Sekiya et al. 2022 ; Lambert et al. 2021). In addition, recent evaluations of the TROPOMI RPRO v2.4 product against MAXDOAS observations still indicate significant biases of TROPOMI NO$_2$ TVCDs of typically -40% over highly polluted areas (Lambert et al., 2023; van Geffen et al., 2022b).**

**The differences between OMI and TROPOMI NO$_2$ TVCDs probably come from differences in their retrieval algorithms, for example from different cloud pressure retrievals or from different albedo datasets, as detailed in Section 2.4.**

L355-360: please explain better what is meant with 'an indirect comparison of OMI and TROPOMI using the CHIMERE CTM as an intermediate'. I am also missing this paragraph's key message and contribution to the subsection.

**OMI and TROMI retrievals do not perfectly coincide in terms of spatio-temporal and vertical representativeness. One can attempt to compare, directly, retrievals from OMI and TROPOMI whose footprints overlap and whose observation time is close, ignoring the differences in terms of vertical sensitivity (AKs). But a more suitable "indirect" comparison can be made by confronting OMI retrievals to CHIMERE in one hand, and TROPOMI retrievals to CHIMERE in the other hand (in both cases: applying the corresponding AKs to CHIMERE), and analysing whether the average signs and levels of differences to CHIMERE are consistent when analysing OMI vs. TROPOMI dataset. This indirect comparison using the CHIMERE CTM as an intermediate benchmark highlights the impact of the different OMI and TROPOMI AKs on the CHIMERE simulations.**

**These explanations have been added in Section 3.1: "Since the OMI and TROPOMI AKs are different, and since the spatio-temporal samplings of the two data sets are also different, a direct comparison between the OMI and TROPOMI datasets could be complex or misleading. The confrontation of the OMI retrievals to CHIMERE in the one hand, and of the TROPOMI retrievals to CHIMERE in the other hand can be used as an indirect but more suitable comparison between the OMI and TROPOMI datasets."**

L394-395: this seems specific to the experiment setup with a missing agricultural soil emission term.

**See our answer to the general comment about the biogenic emissions.**

---

## Author Response (AR2)

**We wish to thank the referee for his/her helpful comments. His/her full comments are copied hereafter in normal black font, and our responses are inserted in blue.**

Many thanks for submitting the revised manuscript, in which my comments were satisfactorily addressed. The revised title is more appropriate, and the clarity of the writing has improved. I also welcome the conceptual figure that has been added.

One of the main outcomes of this study is that the magnitude of the NOx emission reduction in the CIF-CHIMERE inversion system is highly dependent on the choice of super-observation errors. Currently, the authors steer clear of providing any specific recommendations to the community about the choice of the (super-)observation errors, other than the factual statement that choosing "optimistic" errors leads to a better agreement with emission reductions reported in the prior inventory compared to an inversion with conservative uncertainties. I would encourage the authors to provide some additional reflection on this in the conclusions. Would your results provide sufficient evidence to suggest the community to use these optimistic errors? Or would additional, targeted experiments be needed to be able to conclude this, and if so which ones?

Indeed, we consider that further work is needed to provide guidance for the configuration of the observation errors associated to super-observations, via a robust estimate of the relative weight between the NO2 TVCD retrieval errors that are correlated in space and the total retrieval errors, and the characterization of the spatial correlations in the former. Miyazaki et al. (2012) and Boersma et al. (2016) had made a step forward by representing the error correlation between retrievals, based on the consideration that errors in clouds, albedo, a priori profile, and aerosol in retrievals are typically correlated in space, but they acknowledged that the exact number is difficult to estimate. In-depth analyses of the involved variables and evaluation of the uncertainty associated to the retrievals would be valuable.

We have changed sentences in the conclusion : « **Our results, with OMI and TROPOMI data but also with different choices made for the derivation of the error associated with each super-observation, lead to different magnitudes of corrections on NOx anthropogenic emissions. This suggest that more observational constraints and further work would be required to sharpen the European emission estimates. Observational information from future satellite missions such as Sentinel-4 on board geostationary satellites would increase the number of observations for better constraining the NOx emissions in particular for Eastern andNorthern countries. However, even if considering the corresponding increase in the observation sampling and weight, there is a particular need for in depth analysis of the spatial correlations of the error components in the TROPOMI and OMI NO2 TVCD retrievals to support the configuration of the errors on super-observations, as recently highlighted by Rijsdijk et al. (2025).** »